# Structure-Induced Information for Rerooting Levin Tree Search

Jake Tuero [1 2]   Michael Buro [1]   Laurent Orseau [3]   Levi H. S. Lelis [1 2]

## Abstract

Subgoal-based policy tree search, which uses a policy to guide the search, is effective for complex single-agent deterministic problems but often relies on explicit subgoal generation, which can incur substantial overhead, hindering scalability. In this paper, we overcome these limitations by using a learned "rerooter" through the recently introduced $\sqrt{\text{LTS}}$ algorithm. A *rerooter* implicitly decomposes the problem into soft subtasks. While previous work focused on the formal guarantees for given or handcrafted rerooters, in this work, we propose three rerooter designs: (i) a clustering-based rerooter that exploits global state-space structure, (ii) a heuristic-based rerooter that leverages learned cost-to-go estimates, and (iii) a hybrid that combines both signals. Our framework avoids explicitly reconstructing and reasoning over generated subgoals, thereby enabling scalable allocation of search effort with significantly lower computational overhead. Empirically, our rerooting-based methods scale to complex environments where subgoal-based policy tree search fails, and achieve state-of-the-art online training efficiency on the domains tested.

## 1. Introduction

Complex discrete planning problems remain difficult to solve at scale, driving increasing interest in learning-guided search methods. Levin Tree Search (LTS) (Orseau et al., 2018), a tree search algorithm that uses a learned policy to guide the search (a probability distribution over actions), has shown success in addressing these types of problems. A key property of LTS is that it provides an upper bound on the number of search steps required before finding a solution, which depends on the *quality* of the policy. This, in turn,

enables policies to be learned with the explicit objective of minimizing search effort. Policy Guided Heuristic Search (PHS*) (Orseau & Lelis, 2021) extends LTS by combining a learned policy with a learned heuristic function. Orseau & Lelis (2021) provide a similar bound for PHS*, and also show that a policy can be learned while minimizing this bound.

Although LTS and PHS* are theoretically well-founded, they rely primarily on the learned policy and heuristic functions and, without additional structural guidance, can struggle to solve complex problems. A common approach when scaling to complex problem domains, inspired by how humans plan (Botvinick et al., 2009; Donnarumma et al., 2016; Correa et al., 2023), is to decompose the problem into easier subtasks and subgoals. Subgoals help address this limitation by structuring the search and extending its reach beyond what the initial policy can support. Building on this insight, prior work has introduced subgoal-based policy tree search methods, including HIPS-$\varepsilon$ (Kujanpää et al., 2024) and Subgoal-Guided Policy Heuristic Search (SGPS) (Tuero et al., 2025), which generate intermediate target states and condition low-level policies on these generated subgoals to guide the search. By explicitly reasoning over such subgoals, these approaches can improve early-stage exploration and learning efficiency. However, they also introduce additional modeling complexity and computational overhead, as search performance becomes tightly coupled to the quality of subgoal reconstruction and the policies conditioned on them. As we will show, this issue becomes increasingly pronounced as the domain complexity increases.

$\sqrt{\text{LTS}}$ (Orseau et al., 2024), read as "root-LTS", is a policy tree search algorithm that implicitly starts an LTS search at each node in the search tree. The overall search effort is split between each of these searches, and the proportion of time allocated to each is given through a *rerooter*. This mechanism implicitly decomposes the search into subtasks, foregoing the complexity of the subgoal modeling of HIPS-$\varepsilon$ and SGPS that makes those methods costly when scaling to complex domains. Orseau et al. (2024) showed that the bound on the number of node expansions before $\sqrt{\text{LTS}}$ finds a solution can be exponentially better than that of LTS.

In this work, we revisit rerooting as a general mechanism for exploiting structure in the underlying state space in policy

[1]Department of Computing Science, University of Alberta, Edmonton, Canada. [2]Alberta Machine Intelligence Institute (Amii), Edmonton, Canada. [3]Google DeepMind, London, United Kingdom.. Correspondence to: Jake Tuero <tuero@ualberta.ca>.

*Proceedings of the 43rd International Conference on Machine Learning*, Seoul, South Korea. PMLR 306, 2026. Copyright 2026 by the author(s).

tree search and study how to derive rerooting weights in practice to guide search effectively. Within this framework, we present three instantiations. The first two capture complementary structural information: a global approach that induces structure via state-space clusters identified using Leiden clustering (Traag et al., 2019), and a lightweight local approach that derives structure from learned heuristic cost information. We then show how an additive rerooter can take advantage of the strengths each rerooter provides, and instantiate this as a hybrid rerooter of the previous two. In contrast to the subgoal baseline methods, which depend on computationally expensive subgoal generation using high-capacity models (Tuero et al., 2025), our approach does not require learning or invoking separate subgoal networks. Instead, we instantiate rerooters from structure already present in the search tree, with the clustering rerooter running on demand during search. Empirical results show that our rerooters substantially improve online training sample efficiency over non-rerooting baselines. These results establish rerooting as a scalable approach for exploiting structure in search, while avoiding explicit subgoal generation of previous work.

Our contributions can be summarized as follows. We provide automated methods for learning rerooters from the structure of the search tree and show that even lightweight structural signals can achieve strong performance. Finally, we provide a novel theoretical guarantee on the number of node expansions until the first solution node is found by $\sqrt{\text{LTS}}$ using additive rerooters, which highlights how $\sqrt{\text{LTS}}$ can take advantage of the *synergies* of multiple rerooters. The experiments conducted show that our method can scale to complex environments during online training and achieve state-of-the-art training efficiency for methods that use the bootstrap method, whereas previous methods that rely on subgoal reconstruction fall short.

## 2. Preliminaries

Policy tree search algorithms solve single-agent deterministic problems by incrementally constructing a search tree. These problems are represented as a tuple $(\mathcal{S}, \mathcal{A}, T, s_1, \mathcal{S}_g, \ell)$, where $\mathcal{S}$ denotes the state space, $\mathcal{A}$ is a finite actions set, and $T : \mathcal{S} \times \mathcal{A} \to \mathcal{S}$ is a deterministic transition function. The initial state is given by $s_1 \in \mathcal{S}$, and $\mathcal{S}_g$ is the set of goal states. The search problem induces a directed graph $G = (\mathcal{S}, A)$, where an edge $(s, s') \in A$ exists whenever there is an action $a \in \mathcal{A}$ such that $T(s, a) = s'$.

The set of nodes in the search tree is $\mathcal{N}$. The set of children nodes of a node $n$ is $\mathcal{C}(n)$, and its parent is $\text{par}(n)$. All nodes have exactly one parent, except for the root node $n_1$, which corresponds to the initial state $s_1$. The set of ancestors of a node $n$ is $\text{anc}(n)$, and we define $\text{anc}_*(n) = \text{anc}(n) \cup \{n\}$. Similarly, the set of descendants of a node

$n$ is $\text{desc}(n)$, and $\text{desc}_*(n) = \text{desc}(n) \cup \{n\}$. We also use the notation $n' \prec n$ for $n' \in \text{anc}(n)$, and $n' \preceq n$ for $n' \in \text{anc}_*(n)$. The set of nodes representing goal states $\mathcal{S}_g$ is denoted $\mathcal{N}_g$. The search algorithm incurs a loss of $\ell : \mathcal{N} \to (0, \infty]$ for each node expansion. For any node $n$, the *path loss* is defined as $g(n) = \sum_{n' \preceq n} \ell(n')$, *i.e.* the sum of losses from the root to $n$. We assume that all algorithms discussed in this work incur loss $\ell(n) = 1$ for all nodes $n$, resulting in path loss $g(n)$ being equivalent to node depth $d(n) + 1$.

A policy $\pi$ assigns probabilities to child nodes, where $\pi(\cdot|n)$ is a distribution over $\mathcal{C}(n)$. The induced *path probabilities* are defined by $\pi(\underline{n}|\overline{n}) = \pi(\underline{n}|n)\pi(n|\overline{n})$ for any $\overline{n} \preceq n \preceq \underline{n}$, with $\pi(n|n) = 1$. We define $\pi(n|\overline{n}) = 0$ when $\overline{n} \not\preceq n$. For convenience, we denote $\pi(n) = \pi(n|n_1)$. Some search algorithms utilize a *heuristic* $h : \mathcal{N} \to \mathbb{R}_{\geq 0}$ which assigns non-negative values to nodes which estimate the path loss from the current node to a goal node.

### 2.1. Background

**Best-First Search (BFS)**. BFS (Pearl, 1984) expands nodes by increasing *cost*. The search is initialized with the root node in a priority queue. In each search step, the lowest cost node is removed from the queue and is expanded, with the generated nodes added into the queue. A node is not expanded by BFS if its underlying state has previously been expanded. BFS will halt when either there are no nodes left in the queue, a solution is found, or a search budget (in terms of the number of expansions) has been exceeded.

**Levin Tree Search**. Levin Tree Search (LTS) (Orseau et al., 2018) is a BFS algorithm which uses

$$\varphi_{\text{LTS}}(n) = \frac{d(n) + 1}{\pi(n)} \tag{1}$$

as the cost function. LTS is guaranteed to expand no more than $(d(n^*) + 1) / \pi(n^*)$ nodes until the first solution node $n^* \in \mathcal{N}_g$ has been generated (Orseau et al., 2018).

**The $\sqrt{\text{LTS}}$ Algorithm**. $\sqrt{\text{LTS}}$ (Orseau et al., 2024) is a *rerooting* algorithm which implicitly starts an LTS search rooted at every node of the tree. For any node $n$, the base cost function $c_t^r(n)$ represents the cost of $n$ with respect to the instantiated LTS search rooted at node $n_t \prec n$:

$$c_t^r(n) = \sum_{n_t \prec n' \preceq n} \frac{1}{\pi(n'|n_t)}. \tag{2}$$

A *rerooter* assigns a *rerooting weight* to each of these LTS searches, which $\sqrt{\text{LTS}}$ uses to split the overall search effort between these searches proportional to their assigned weight. Like LTS, $\sqrt{\text{LTS}}$ is a BFS algorithm using the cost function

$$c^r(n) = \min_{n_t \prec n} \frac{1}{w_t} c_t^r(n), \tag{3}$$

where $w_t \geq 0$ is the rerooting weight for the LTS search anchored at the ancestor $n_t$ of $n$.

Orseau et al. (2024) also provide theoretical guarantees on the number of node expansions required until the first solution node has been found, which depends on the quality of the policy and the rerooter. Suppose $\sqrt{\text{LTS}}$ finds the solution node $n^*$ at some step $T$. A *subtask decomposition* of the path from $n_1$ to $n^*$ is a subset of the ancestors of $n^*$, which must include both $n_1$ and $n^*$, viewed as subtask boundaries. For example, for the game of Sokoban, one subtask decomposition is all the ancestors where the agent has pushed a box on a goal spot. Let $\mathcal{D}(n^*)$ be the set of all such subtasks decompositions of $n^*$. For $D \in \mathcal{D}(n^*)$, the selected ancestors of $n^*$ are expanded at steps $T_1, T_2, \ldots, T_{|D|}$, where necessarily $T_1 = 1$ and $T_{|D|} = T$. Then the number of search steps $T$ that $\sqrt{\text{LTS}}$ takes to visit $n^*$ is bounded by (Orseau et al., 2024, Corollary 12, adapted)

$$T \leq 1 + \min_{D \in \mathcal{D}(n^*)} \max_{i < |D|} \frac{w_{<T}}{w_{T_i}} c_{T_i}^r(n_{T_{i+1}}), \qquad (4)$$

where $w_{<T} = \sum_{j<T} w_t$ is the cumulative rerooting weight of the nodes expanded up to step $T$ (excluded), $w_{T_i}/w_{<T}$ is the time share allocated to the LTS instance started at the $i$th (shallowest) ancestor of $n^*$ in the subtask decomposition, $c_{T_i}^r(n_{T_{i+1}})$ is the bound on the number of search steps this LTS instance takes to reach the next subtask boundary $n_{T_{i+1}}$, and $\max_{i<D}$ corresponds to the most 'difficult' subtask in the decomposition. This subtask decomposition can allow $\sqrt{\text{LTS}}$ to expand exponentially fewer nodes than LTS; see (Orseau et al., 2024) for a detailed description of $\sqrt{\text{LTS}}$. See Appendix J for an example.

## 2.2. Problem Definition

Our objective formulation is the same as given in Orseau & Lelis (2021) and Tuero et al. (2025), which is to solve a set $K$ of problem instances as *quickly* as possible. We use the *total search loss* as our objective metric. The *search loss* $L(S, n)$ of algorithm $S$ is the sum of losses $\ell(n')$ incurred for each node $n'$ expanded up to and including $n$. Here, $n$ could either be a solution node in $\mathcal{N}_g$ or the last node expanded before a given budget is exceeded. The *total search loss* $\sum_{k \in K} L(S, n_k)$ is the sum of individual search losses algorithm $S$ incurs on problem $k$. We assume $\ell(n) = 1$ everywhere, thus minimizing the search loss corresponds to minimizing the total number of expansions.

## 3. Structure-Induced Rerooters

In this section, we describe how rerooting can be instantiated in practice to exploit structural information during policy tree search. Orseau et al. (2024) focused on analyzing the formal properties of $\sqrt{\text{LTS}}$ for a *given* rerooter, and left open the question of how to define or learn the rerooter. We

address this gap by presenting two complementary rerooters that derive rerooting weights from signals available during search: a clustering-based rerooter that captures global structure in the state space, and a heuristic-based rerooter that leverages local cost-to-go information. Together, these designs illustrate how rerooting can be learned or instantiated automatically, without the complexities of explicit subgoal reconstruction, and provide a practical foundation for scalable rerooting in complex domains.

### 3.1. Global Structure–Induced Rerooting

**Motivation**. Progress in many planning problems is characterized by transitions between distinct regions of the state space. When search reaches a new region, it is often beneficial to concentrate effort there; when it keeps expanding within the same region without making progress, effort should be redirected elsewhere in the tree. Such transitions may correspond to events like entering a new room or acquiring a key that unlocks additional states. To capture this behavior, the global rerooter derives rerooting weights from coarse state-space structure by clustering states based on connectivity. It then allocates effort across clusters rather than individual nodes, reflecting global organization while relying only on state observations and a Boolean goal test, without explicit subgoal reconstruction during search.

**Global Structure**. A key part of our method is the Leiden clustering algorithm (Traag et al., 2019), which is an improvement over the Louvain clustering algorithm (Blondel et al., 2008), both in terms of runtime complexity and cluster quality. The Leiden algorithm is an iterative algorithm that creates a hierarchy of graphs $(G_1, \ldots, G_N)$ where $G_{i+1}$ is a clustering of graph $G_i$. In each iteration $i$, a hierarchical cluster graph $G_{i+1} = (V_{i+1}, E_{i+1})$ is created from $G_i = (V_i, E_i)$, where $V_{i+1}$ is a partition of $V_i$ with each $v \in V_{i+1}$ being a part in the partition, and $E_{i+1}$ containing edges $(v_{i+1}, v'_{i+1})$ for $v_{i+1}, v'_{i+1} \in V_{i+1}$ if there is a $u \in v_{i+1}$ and $v \in v'_{i+1}$ such that $(u, v) \in E_i$. The process continues until either there is no progress made (i.e., $G_{i+1} = G_i$) or $G_{i+1}$ contains a single node.

Our first rerooter, which we denote $\sqrt{\text{LTS}}$-L, uses the Leiden clustering algorithm to find structures in the induced subgraph of the state space. During the search, the induced subgraph of the underlying state space is constructed iteratively as the tree is built. When the search generates a tree-node representing an unseen state, it adds a graph-node to the induced subgraph with an edge linking to the graph-node represented by the parent tree-node. When the search generates a tree-node representing a previously seen state, it adds an edge to the induced subgraph linking the graph-node represented by the parent tree-node to the graph-node representing the state of the generated tree-node.

The Leiden algorithm creates cluster graphs at increasing

levels of the hierarchy by maximizing the *modularity*, a metric that quantifies the relation between the edge connectivity within a cluster and the connectivity between clusters. High modularity results in dense clusters with many connections within each cluster, and sparse connections between clusters. Evans & Şimşek (2023) found that the Louvain algorithm using this modularity metric finds meaningful structures in the state spaces they studied, such as two neighboring clusters corresponding to states of two adjacent rooms in a gridworld environment.

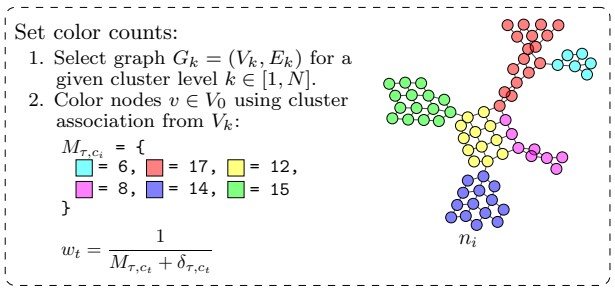

*Figure 1.* $\sqrt{\text{LTS}}$-L **Rerooter**. Each node in the tree is assigned a color corresponding to the cluster it is associated with in the cluster graph at level $k$ of the hierarchy. The rerooting weight for the expanded node is then computed using the color count map.

**Rerooting Weight**. The $\sqrt{\text{LTS}}$-L rerooter is visually depicted in Figure 1. We invoke the Leiden algorithm at search steps $\gamma^i$ for $i \in \mathbb{Z}_{\geq 0}$ on the incrementally induced subgraph $G_0$, with hyperparameter $\gamma > 0$. From the resulting hierarchy, we select a cluster graph $G_k = (V_k, E_k)$ at hierarchy level $k$ (a hyperparameter), where each node $U \in V_k$ corresponds to a set of nodes in $G_0$ (and thus to a set of search-tree nodes). We assign each tree node $n$ a color $c \in \{1, \ldots, |V_k|\}$ indicating the node $U$ that contains it; we refer to this assignment as a *coloring*. Colors may change across invocations: a node $n$ can have color $i$ on the $(\tau)$th invocation, and color $j \neq i$ on the $(\tau + 1)$th invocation.

At search step $t$, the Leiden algorithm has been invoked $\tau = \lfloor \log_\gamma(t) \rfloor$ times, resulting in $\tau$ colorings, with the most recent coloring occurring on search step $t' = \lfloor \gamma^\tau \rfloor \leq t$. For the search steps between $t'$ and $t$, some nodes may be expanded which do not have a color, as they were not present in the induced subgraph when the most recent $(\tau)$th coloring occurred. For these nodes, we assume that they belong to the same clustering as their parent and are thus given their parent's color. Using a proxy value to defer expensive computations has been used previously in heuristic search, such as when computing the edge cost (Narayanan & Likhachev, 2017) or the heuristic value (Karpas et al., 2018) is costly.

Let $M_{\tau,c}$ be the number of nodes with color $c$ from the $(\tau)$th

coloring, and

$$\delta_{\tau,c} = |\{\lfloor \gamma^\tau \rfloor < \ell \leq t : c_\ell = c\}| \qquad (5)$$

be the count of nodes expanded since step $\lfloor \gamma^\tau \rfloor$ which also have color $c$. Then, $\sqrt{\text{LTS}}$-L assigns the rerooting weight $w_t$ for the node $n_t$ expanded at search step $t$ as

$$w_t = \frac{1}{M_{\tau,c_t} + \delta_{\tau,c_t}} \qquad (6)$$

where $c_t$ is the color associated with $n_t$. The value of $M_{\tau,c_t} + \delta_{\tau,c_t}$ approximates the size of the cluster $c_t$. Note that the cluster size is only approximate because the search will potentially generate and expand nodes that were not in $G_0$ during the last execution of the clustering algorithm.

This gives us the desirable property that a low color count results in a larger weight $w_t$ and lower search cost. Similarly, a higher color count results in a smaller weight $w_t$, and higher cost. As the search expands more nodes with color $c$, the count increases and thus subsequent nodes expanded from the same cluster receive lower weights and higher costs. See Algorithm 1 in Appendix A for its pseudocode.

**Runtime Complexity**. While the induced subgraph of the state space is built incrementally, Leiden cluster graphs are recomputed on the most recent graph under a geometric update schedule to limit runtime overhead. Although cluster assignments (and sizes) may change across updates, we keep the rerooting weights of previously expanded nodes as fixed: retroactively updating them would make the search tree internally inconsistent, since an ancestor could induce different costs for different descendants depending on when they were generated. Although repeated clustering may appear expensive, we show that $\sqrt{\text{LTS}}$-L's runtime matches BFS up to constant factors. Unlike prior subgoal-based approaches, it avoids queries to compute-intensive models such as subgoal generators (Tuero et al., 2025).

**Theorem 3.1.** *Let $N$ be the number of node expansions, $D$ the depth of the max-depth node after $N$ expansions, $k$ the cluster hierarchy level, $G = (V, E)$ the underlying state space graph for environment domain. Assume the environment has a uniform branching factor $b < \infty$. If Leiden clustering is invoked at search steps following a geometric schedule with factor $\gamma > 1 + 1/\epsilon$ for some constant $\epsilon > 0$, then $\sqrt{\text{LTS}}$-L has $O(bN \log N + DN)$ time complexity.*

The proof is in Appendix B. The uniform branching-factor assumption is standard in tree-search–based planning and control, where each state has a bounded number of successor states determined by the action set.

### 3.2. Local Structure–Induced Rerooting

**Motivation**. Global structural cues are not always needed for effective rerooting: local information at individual nodes

can distinguish between otherwise similar regions of the search tree. For example, if the root has two structurally identical subtrees, a rerooter based solely on structural expansion history would allocate equal effort to both; yet if one subtree is estimated to be closer to the goal, it should be prioritized. We therefore consider a local rerooter that leverages heuristic cost-to-go estimates to assign weights reflecting the local geometry of the search space.

**Rerooting Weight**. The rerooter, which we denote as $\sqrt{\text{LTS}}$-H, uses the heuristic value of a node to define its rerooting weight directly. The rerooting weight is given by

$$w_1 = 1; \quad w_t = \exp\left(-\alpha \frac{h(n_t)}{h(n_1)}\right), \quad (7)$$

where $\alpha > 0$ is a constant constant controlling how strongly rerooting effort is concentrated on nodes with low heuristic value, and $h(n_t)$ is the heuristic value corresponding to node $n_t$.

This choice is natural for several reasons. First, it is monotone in the heuristic: if $h(n_i) < h(n_j)$, then $w_i > w_j$. Thus, nodes estimated to be closer to the goal receive a larger rerooting weight and therefore induce a lower rerooted search cost. Normalizing by the root heuristic $h(n_1)$ makes the rerooting weight depend on relative rather than absolute heuristic scale. This is useful when different environments, or different instances within the same environment, exhibit substantially different effective solution-length scales. Second, the exponential map is smooth and strictly positive. Even nodes with relatively poor heuristic values retain non-zero weight, which is desirable when the heuristic is imperfect and should influence, rather than fully determine, the rerooting decision. Third, the parameter $\alpha$ acts as an inverse-temperature parameter. When $\alpha$ is small, the weights are more similar across nodes, so the rerooter behaves conservatively. As $\alpha$ increases, the weight mass becomes more concentrated on nodes whose heuristic values are small relative to the root, causing rerooting to focus more aggressively on regions that appear to have made substantial progress toward a goal.

A useful consequence of this normalization is that this rerooting formulation is invariant to multiplicative rescaling of the heuristic. The exponential form also gives the rerooter a useful normalized interpretation. For any set $I$ of candidate rerooting points,

$$\frac{w_t}{\sum_{i \in I} w_i} = \frac{\exp(-\alpha h(n_t)/h(n_1))}{\sum_{i \in I} \exp(-\alpha h(n_i)/h(n_1))},$$

which is precisely a softmax over the scores $-\alpha h(n_i)/h(n_1)$. Thus, the rerooter distributes search effort smoothly across candidate local roots, with larger shares assigned to nodes whose heuristic values are small relative to the root. This is well aligned with the role of rerooting weights in $\sqrt{\text{LTS}}$, where the weights determine how computation is shared among competing local searches. See Algorithm 2 in Appendix A for its pseudocode.

### 3.3. Hybrid Structure–Induced Rerooting

**Motivation**. While our global and local rerooters are each useful in isolation, they capture complementary aspects of the search problem and exhibit distinct limitations. The heuristic-based rerooter provides a lightweight goal-directed signal, but on its own it depends entirely on local cost-to-go estimates and can therefore be sensitive to heuristic noise or miscalibration. In particular, if the heuristic is only weakly informative in some region of the tree, the rerooter may overemphasize nodes whose low heuristic values do not correspond to meaningful structural progress. By contrast, the clustering-based rerooter relies on coarse connectivity in the induced state-space graph and is agnostic to direct goal proximity, making it less responsive to local progress toward a solution. Consequently, it may spread search effort across regions that are structurally distinct yet equally far from the goal.

These two rerooters are therefore naturally complementary. The global rerooter captures coarse structural bottlenecks and allocates effort across broader regions of the search space, while the local rerooter refines this allocation using heuristic information that reflects estimated progress within those regions. This motivates a hybrid rerooter that combines both signals additively.

**Rerooter Weight**. The hybrid rerooter combines these signals by using global structure to set a stable coarse allocation of effort, and heuristic estimates to refine priorities within each region. This yields a coarse-to-fine rerooting rule that is less sensitive to heuristic miscalibration while retaining goal-directed adaptivity across domains. We denote this hybrid rerooter as $\sqrt{\text{LTS}}$-LH, and use the combined rerooting weights given by

$$w_t = u_a \frac{1}{M_{\tau,c_t} + \delta_{\tau,c_t}} + u_b \exp\left(-\alpha \frac{h(n_t)}{h(n_1)}\right), \quad (8)$$

where $c_t$ is the color associated with $n_t$, $M_{\tau,c_t}$ and $\delta_{\tau,c_t}$ follow the same definition as in Equation 6, and $u_a$ and $u_b$ are mixing coefficients. Unless stated otherwise, $\sqrt{\text{LTS}}$ uses $u_a = u_b = 1$. As before, we set $w_1 = 1$ for the root node. See Algorithm 3 in Appendix A for its pseudocode.

**Additive Rerooter Guarantees**. The hybrid rerooter in Equation 8 is not only a pragmatic way to blend rerooters, it also has a principled effect on how $\sqrt{\text{LTS}}$ allocates effort across subtasks. Intuitively, adding rerooters allows the search to fall back on whichever signal is informative in a given region of the tree, provided the combination remains balanced so that one component does not dominate the over-

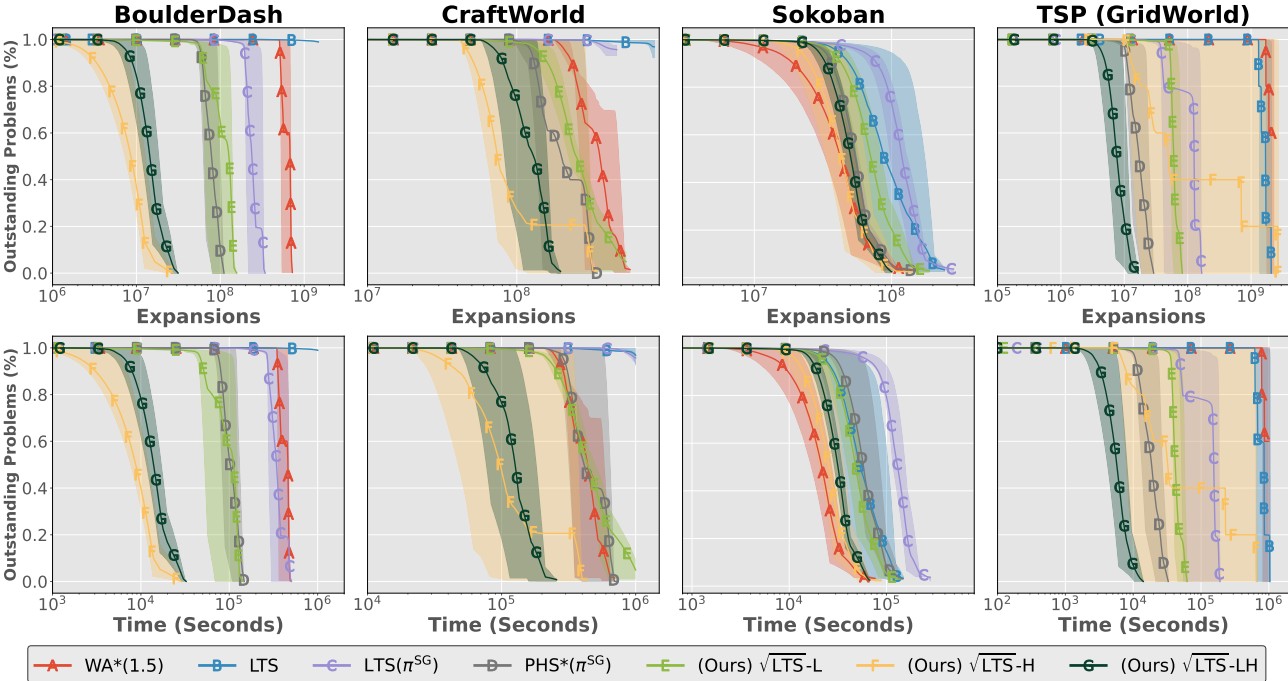

*Figure 2.* Average online training loss with respect to expansions (top) and time (bottom), in log-scale. Shaded regions show the minimum and maximum. Time measures the sum an algorithm spends on each problem across all threads used during training.

all time allocation. The following result makes this precise by extending the standard subtask-decomposition bound.

**Theorem 3.2.** *Let $w_a$ and $w_b$ be two rerooters, with relative weights $u_a \geq 0$ and $u_b \geq 0$. Let $w = u_a w_a + u_b w_b$ be the rerooter used by $\sqrt{\text{LTS}}$. When visiting the node $n_T$ at step $T$, for any $C \geq 1$ satisfying $\frac{1}{C} \leq \frac{u_a w_{a,<T}}{u_b w_{b,<T}} \leq C$, then the number of node visits is bounded by*

$$T \leq 1 + (C+1) \times$$
$$\min_{D \in \mathcal{D}(n^*)} \max_{i < |D|} \min \left\{ \frac{w_{a,<T}}{w_{a,T_i}}, \frac{w_{b,<T}}{w_{b,T_i}} \right\} c^r_{T_i}(n_{T_{i+1}}).$$

The proof is in Appendix J. The constant $C \geq 1$ can be controlled by adjusting the weights $u_a$ and $u_b$.

# 4. Experiments

The goal of our experimental evaluation is twofold. First, we aim to establish rerooting as a flexible and scalable abstraction for exploiting structural information in policy tree search, avoiding explicit subgoal generation and reasoning while retaining much of their benefit through simple structural signals. Second, we evaluate the practical impact of rerooting on learning efficiency by measuring improvements in online training sample efficiency relative to non-rerooted baselines across a range of domains.

## 4.1. Baselines

We compare our methods against LTS (Orseau et al., 2018), as at the time of this writing, there are no comprehensive empirical evaluations of $\sqrt{\text{LTS}}$ variants against LTS. We also compare against LTS($\pi^{\text{SG}}$) and PHS*($\pi^{\text{SG}}$) (Tuero et al., 2025), two SGPS instantiations that can be trained online and use the same clustering method as $\sqrt{\text{LTS}}$-L. SGPS is a key baseline because it (i) shares the same clustering component, (ii) generates discrete subgoals via a VQ-VAE (Van Den Oord et al., 2017) whereas our approach uses soft subtasks without explicit subgoal generation, and (iii) achieved state-of-the-art results on the domains we consider. Finally, we include Weighted A* (WA*) (Pohl, 1970), a non-policy method that can be trained online via the bootstrap process. We use a weight of 1.5 for WA*, which performs well in similar settings (Orseau & Lelis, 2021; Tuero et al., 2025). We omit HIPS-$\varepsilon$ because it requires a solution dataset which is unavailable for the complex environment studied, and prior work found LTS($\pi^{\text{SG}}$) and PHS*($\pi^{\text{SG}}$) outperformed it (Tuero et al., 2025).

## 4.2. Environment Domains

**BoulderDash**: The agent collects a number of diamonds to unlock the exit. Some diamonds are locked in rooms that require a key. The environment contains dirt cells, which disappear when the agent walks over them, resulting in complex state observations and a large search space. Tuero

et al. (2025) provided a *standard* problem set and a *hard* problem set, and we use the more challenging hard problems in our analysis.

**CraftWorld**: The agent collects raw materials and interacts with workbenches and furnaces to craft intermediary items, which can further be crafted into final products (Andreas et al., 2017). The environment can be in a deadlocked state by crafting an incorrect item, since items are consumed once used in a recipe. Similar to BoulderDash, we use the *hard* problems from Tuero et al. (2025).

**Sokoban**: The agent must push boxes into designated goal locations, avoiding deadlocks due to boxes getting stuck along walls. Sokoban is PSPACE-hard (Culberson, 1997), and we use the first 50,000 training problems and 1,000 test problems from Boxoban (Guez et al., 2018).

**Traveling Salesman Problem (TSP)**: A gridworld version of the TSP where the agent must visit specified city locations, then return to the starting location. To increase the difficulty of this domain, we use a modified version in which the agent deadlocks if it revisits a city other than the start city. This prevents trivial solutions such as blindly visiting each city without planning the steps in between cities.

### 4.3. Training and Testing Procedure

All methods use Bootstrap training (Arfaee et al., 2011). We randomly initialize neural networks for the policy and/or heuristic, then run each search algorithm on a subset of training problems with an initial expansion budget using the current policy/heuristic. Algorithms update their policy/heuristic from solution trajectories on solved problems. If an algorithm solves no new problems in a sweep over the training set, we increase the expansion budget and repeat the sweep. After each sweep, we evaluate on a separate validation set under the current budget; training ends once at least 95% of validation problems are solved. To handle methods which struggle to make progress, we impose a maximum training time of 1,000,000 seconds (approximately 11.5 CPU-days), stopping once this limit is reached. Reported time is the sum of time each algorithm uses on each problem, over all the threads used during training. Additional details are in Appendix C and Appendix D.

Each domain uses 10,000 training problems and 1,000 validation problems, except Sokoban (49,000 training and 1,000 validation). We repeat training over 5 seeds, which determine network initialization and the train/validation splits. For training-efficiency plots, we report the min/mean/max number of outstanding problems versus expansions and wall-clock time. For final evaluation, we test on a held-out test set with a budget of 512,000 node expansions and report (averaged over seeds) problems solved, expansions-to-solution, solution length, and wall-clock time.

*Table 1.* Test results averaged over all seeds. Time measures the sum an algorithm spends on each problem across all threads used during training, in seconds.

| Algorithm | Solved | Expansions | Length | Time |
|---|---|---|---|---|
| **BoulderDash** | | | | |
| WA*(1.5) | 100 | 480.52 | **68.56** | 0.73 |
| LTS | 10 | 195,451.08 | 68.92 | 119.86 |
| LTS($\pi^{SG}$) | 100 | 202.38 | 83.09 | 1.53 |
| PHS*($\pi^{SG}$) | 100 | 359.86 | 86.61 | 2.70 |
| $\sqrt{\text{LTS}}$-L | 100 | 131.18 | 71.63 | 0.61 |
| $\sqrt{\text{LTS}}$-H | 100 | **92.37** | 69.47 | 0.60 |
| $\sqrt{\text{LTS}}$-LH | 100 | 92.68 | 69.55 | 0.58 |
| **CraftWorld** | | | | |
| WA*(1.5) | 100 | 1,888.35 | **122.16** | 3.62 |
| LTS | 100 | 306,224.22 | 142.37 | 373.44 |
| LTS($\pi^{SG}$) | 100 | 345,096.42 | 170.99 | 869.89 |
| PHS*($\pi^{SG}$) | 100 | 1,413.00 | 126.01 | 8.67 |
| $\sqrt{\text{LTS}}$-L | 100 | 8,802.62 | 183.74 | 44.15 |
| $\sqrt{\text{LTS}}$-H | 100 | 2,514.62 | 125.11 | 6.84 |
| $\sqrt{\text{LTS}}$-LH | 100 | **1,347.52** | 134.44 | 4.59 |
| **Sokoban** | | | | |
| WA*(1.5) | 1,000 | **1,201.03** | **32.91** | 0.51 |
| LTS | 1,000 | 1,914.06 | 36.36 | 0.76 |
| LTS($\pi^{SG}$) | 1,000 | 2,010.88 | 35.82 | 1.98 |
| PHS*($\pi^{SG}$) | 1,000 | 1,630.60 | 34.66 | 1.56 |
| $\sqrt{\text{LTS}}$-L | 1,000 | 2,579.13 | 39.49 | 1.53 |
| $\sqrt{\text{LTS}}$-H | 1,000 | 2,626.82 | 34.76 | 1.60 |
| $\sqrt{\text{LTS}}$-LH | 1,000 | 1,736.03 | 36.37 | 1.10 |
| **TSP (GridWorld)** | | | | |
| WA*(1.5) | 100 | 180,381.42 | **35.40** | 62.75 |
| LTS | 100 | 216.61 | 39.43 | 0.33 |
| LTS($\pi^{SG}$) | 100 | 76.99 | 36.40 | 0.69 |
| PHS*($\pi^{SG}$) | 100 | **46.31** | 36.39 | 0.49 |
| $\sqrt{\text{LTS}}$-L | 100 | 84.35 | 36.69 | 0.42 |
| $\sqrt{\text{LTS}}$-H | 100 | 55.44 | 35.92 | 0.37 |
| $\sqrt{\text{LTS}}$-LH | 100 | 56.12 | 36.16 | 0.38 |

### 4.4. Results

**Search Efficiency**. Across all domains (Figure 2), our hybrid rerooter substantially improves the bootstrap training efficiency over LTS and prior subgoal-generation approaches; comparisons among the $\sqrt{\text{LTS}}$ rerooter variants are discussed separately below. The results highlight the value of combining the rerooters: In BoulderDash, $\sqrt{\text{LTS}}$-H is effective and $\sqrt{\text{LTS}}$-LH matches its performance. The remaining domains, Sokoban, CraftWorld, and TSP, all contain deadlock structure, where purely heuristic guidance can be less stable. In these domains, $\sqrt{\text{LTS}}$-H remains competitive but exhibits greater variability, suggesting that the learned heuristic can sometimes overcommit search effort to misleading regions of the tree during bootstrap training. This effect is most pronounced in CraftWorld and TSP; in these environments, WA* also requires substantially more time to complete training. By contrast, $\sqrt{\text{LTS}}$-LH is both the strongest and most stable method in these domains, indicating that the hybrid rerooter benefits from the heuristic

*Table 2.* Training results averaged over increasing complexity BoulderDash environments. NPS stands for nodes per second, and time is measured in hours.

| ALGORITHM | EXPANSIONS | TIME | SOLVED | NPS |
|---|---|---|---|---|
| BOULDERDASH (10%) | | | | |
| PHS*($\pi^{SG}$) | 30,686,786 | 10.63 | 10,000 | 802.1 |
| $\sqrt{LTS}$-L | 28,455,464 | 8.36 | 9,998 | 946.0 |
| $\sqrt{LTS}$-H | 17,692,073 | 5.00 | 9,999 | 981.9 |
| $\sqrt{LTS}$-LH | 18,654,789 | 5.36 | 10,000 | 967.4 |
| BOULDERDASH (20%) | | | | |
| PHS*($\pi^{SG}$) | 299,672,516 | 137.12 | 10,000 | 607.1 |
| $\sqrt{LTS}$-L | 65,307,017 | 20.38 | 9,997 | 889.9 |
| $\sqrt{LTS}$-H | 19,866,165 | 5.99 | 9,997 | 921.4 |
| $\sqrt{LTS}$-LH | 25,971,412 | 7.70 | 9,992 | 936.6 |
| BOULDERDASH (30%) | | | | |
| PHS*($\pi^{SG}$) | 429,407,720 | 278.23 | 11 | 428.7 |
| $\sqrt{LTS}$-L | 225,438,691 | 72.51 | 9,964 | 863.7 |
| $\sqrt{LTS}$-H | 27,965,657 | 8.37 | 9,996 | 928.4 |
| $\sqrt{LTS}$-LH | 57,780,467 | 16.50 | 10,000 | 972.5 |
| BOULDERDASH (40%) | | | | |
| PHS*($\pi^{SG}$) | — | — | — | — |
| $\sqrt{LTS}$-L | 458,508,999 | 153.23 | 9,903 | 831.20 |
| $\sqrt{LTS}$-H | 38,524,592 | 11.94 | 9,994 | 896.58 |
| $\sqrt{LTS}$-LH | 75,672,897 | 22.49 | 9,995 | 934.79 |
| BOULDERDASH (50%) | | | | |
| PHS*($\pi^{SG}$) | — | — | — | — |
| $\sqrt{LTS}$-L | 895,423,066 | 317.41 | 605 | 783.6 |
| $\sqrt{LTS}$-H | 72,358,349 | 22.86 | 9,995 | 879.1 |
| $\sqrt{LTS}$-LH | 97,117,682 | 30.13 | 9,990 | 895.2 |

signal when it is useful while retaining the structural robustness of the clustering rerooter $\sqrt{LTS}$-L when the heuristic is unreliable. $\sqrt{LTS}$-L substantially outperforms LTS($\pi^{SG}$) despite both using the same clustering technique and neither using heuristics. The cost-to-go rerooter $\sqrt{LTS}$-H outperforms WA* in every domain except Sokoban. Runtime measurements further support Theorem 3.1: while all methods share BFS-style asymptotic complexity, our rerooters incur the lowest overhead in practice. Together, these results isolate the effect of rerooting, showing that gains come from how structural information is exploited, not clustering alone, and that our rerooters scale reliably across diverse domains. For the full table of results from Figure 2, see Table 8 in Appendix K.

**Test Results**. To assess whether the training speedup degrades solution quality, we evaluate the trained models on held-out test problems (100 per domain; 1,000 for Sokoban; Table 1). Methods that failed to complete training perform substantially worse, since their policies and heuristics remain largely uninformed. Despite faster training, our rerooting methods achieve comparable test performance to prior approaches, indicating that the efficiency gains do not come at the expense of learned policy quality. More-

over, $\sqrt{LTS}$-LH consistently outperforms the clustering-only rerooter $\sqrt{LTS}$-L at test time, suggesting that combining global structure with local heuristic information improves generalization. In TSP and Sokoban, our rerooting methods require more node expansions than PHS*($\pi^{SG}$) in both and WA* in Sokoban. We suspect this reflects weaker state-space clustering structure in these environments, and in Appendix I we perform an analysis to verify this conjecture.

**Environment Domain Complexity Scaling**. To test whether subgoal-based policy tree search methods like PHS*($\pi^{SG}$) scale poorly as problem complexity increases, we run a training-efficiency experiment on BoulderDash while varying the fraction of dirt tiles which are task irrelevant elements that substantially enlarge the state-space.

As complexity increases, PHS*($\pi^{SG}$) times out and makes no progress once levels contain 30% or more dirt elements (Table 2). We attribute this to its reliance on reconstructing grounded future states as subgoals: additional dirt shifts the reconstruction objective toward visually dense but task-irrelevant structure, diverting capacity from critical elements such as keys or diamonds. Although LTS($\pi^{SG}$) and PHS*($\pi^{SG}$) are more robust than earlier subgoal-based methods like HIPS-$\varepsilon$ to invalid subgoal predictions (Tuero et al., 2025), their low-level policies are still conditioned on generated subgoals, making learning increasingly brittle as state complexity grows.

In contrast, $\sqrt{LTS}$-L scales better by avoiding explicit subgoal reconstruction, though it eventually reaches a complexity threshold where training does not complete. $\sqrt{LTS}$-H and $\sqrt{LTS}$-LH remain robust across all tested levels, completing training even in the hardest settings with comparable performance. For an analysis for why $\sqrt{LTS}$-L does not scale as well as compared to $\sqrt{LTS}$-H and $\sqrt{LTS}$-LH, see Appendix I for ablations on how the percentage of dirt elements affects the quality of the clusters found. Overall, implicitly representing subtasks via rerooting scales substantially better than conditioning policies on explicitly generated subgoals.

## 5. Related Work

**Subgoal Search**. Prior work has represented these through fixed-length subtasks (Czechowski et al., 2021), multi-model approaches that represent different subtask lengths by training a separate fixed-horizon model for each length (Zawalski et al., 2022), and performing a high-level search in subgoal-space using a subgoal generator model for varying-length subgoals (Kujanpää et al., 2023). While these approaches show promising results, they all suffer from a lack of completeness, *i.e.*, neither are guaranteed to find a solution even if one exists. HIPS-$\varepsilon$ (Kujanpää et al., 2024) added completeness guarantees by augmenting the search

space to include both subgoals and atomic actions. Subgoal Guided Policy Heuristic Search (SGPS) (Tuero et al., 2025) generates subgoals like HIPS-$\varepsilon$ but performs search over atomic actions. The generated subgoals are used to condition low-level policies, which are mixed with a high-level subgoal policy to produce a single policy usable by LTS or PHS*. A novelty of SGPS is that it can be trained online using the data from the search tree, even when a budget for the search causes early termination with no solution found. Both HIPS-$\varepsilon$ and SGPS are reliant on generated subgoals, and the performance of the search becomes tightly coupled to the quality of subgoal reconstruction and the policies conditioned on them. As shown in our experiments, this becomes an issue when scaling to complex domains.

**State-Space Structure for Search Control**. A complementary line of work leverages structural regularities in the state space, often via partitions or region abstractions, to guide how search allocates effort. Cartesian Counterexample-Guided Abstraction Refinement (Clarke et al., 2000; Seipp & Helmert, 2013) guides search in cost-optimal planning through iteratively refining abstractions. Recent work improves refinement strategies (Speck & Seipp, 2022) and ways of combining multiple abstractions for stronger heuristics (Salerno et al., 2025). In the options framework (Sutton et al., 1999), methods exploit connectivity and bottleneck structure to create temporally extended actions that reduce planning complexity. This includes using entropy to discover skills that capture key transition patterns in the state-space (Zeng et al., 2023; 2025), and state-space clustering (Agostinelli et al., 2019; Ramesh et al., 2019). In contrast, we exploit structure only to steer the search effort through rerooting, without constructing abstractions for heuristic computation or learning temporally extended actions.

## 6. Conclusions

In this work, we examined rerooting as a general mechanism for exploiting structure in policy tree search and demonstrated its effectiveness across a range of instantiations. We introduced two complementary rerooters, and showed how they can be combined to take advantage of the benefits each provides while offsetting each of their downsides. Our rerooter achieves state-of-the-art performance in online training efficiency when using the bootstrap approach, foregoing the need to rely on costly subgoal generation models and state reconstruction. Our results show that rerooting enables scalable and efficient search control by implicitly representing subtasks through weighted root selection rather than explicit subgoal reconstruction.

Perhaps surprisingly, a simple heuristic-based rerooter often outperforms a clustering-based alternative despite relying on substantially less structural information, highlighting that lightweight local signals can be sufficient to guide effective

rerooting. At the same time, the heuristic rerooter is not without limitations: under certain conditions, normalization effects can cause rerooting weights to collapse, leading to degraded performance. The hybrid rerooter addresses this tradeoff by combining global structural guidance with local heuristic information, achieving performance close to the heuristic-based approach while being robust across domains.

Together, these findings position rerooting as a flexible and practical abstraction, and suggest that combining complementary structural signals offers a reliable path toward scalable policy tree search. We have demonstrated that $\sqrt{\text{LTS}}$ with a simple transformation of the classical heuristic *cost-to-go* can be very efficient. We have also successfully combined two rerooters. This opens the door to incorporating different types of side information to define more capable rerooters.

## Acknowledgements

This research was supported by Canada's NSERC and the CIFAR AI Chairs program. This research was enabled in part by support provided by the Digital Research Alliance of Canada. The authors thank the anonymous reviewers for valuable feedback on this work.

## Impact Statement

This work advances the field of machine learning by developing practical rerooting mechanisms that improve how policy-guided tree search allocates effort in hard single-agent problems. Our contributions are primarily technical and evaluated in game-like domains, but the core ideas may transfer to broader planning and decision-making settings over time. We do not foresee immediate societal harms, but any real-world use should follow established ethical and safety practices.

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

# A. Complete $\sqrt{\text{LTS}}$ Algorithm with Leiden and Heuristic Rerooters

Below details how the $\sqrt{\text{LTS}}$-L, $\sqrt{\text{LTS}}$-H, and $\sqrt{\text{LTS}}$-LH rerooters are integrated into $\sqrt{\text{LTS}}$ search. The policy $\pi_\theta$ and heuristic $h_\omega$ are parameterized by neural networks $\theta$ and $\omega$ respectively.

---

**Algorithm 1** $\sqrt{\text{LTS}}$-L

---

1: **Input:** Root node $n_1$, budget $b > 0$, policy $\pi_\theta$, graph update factor $\gamma > 1$, cluster level $k$
2: **Output:** Solution node, or status of failed search
3: $Q_1 \leftarrow \{n_1\}$
4: $V_0 \leftarrow \{n_1\}$, $E_0 \leftarrow \{\}$
5: $t \leftarrow 1$, $r \leftarrow 1$, $\tau \leftarrow 1$
6: **while** $Q_t \neq \{\}$ **and** $t \leq b$ **do**

7:     $n_t = \underset{n \in Q_t}{\arg\min} \; \underset{n_i \prec n}{\min} \; \dfrac{1}{w_i} \left( \sum_{n_i \prec n' \preceq n} \dfrac{1}{\pi_\theta(n'|n_i)} \right)$

8:     **if** is_solution($n_t$) **then**
9:         **return** $n_t$
10:     // Find node's color
11:     **if** is_root($n_t$) **then**
12:         $w_t \leftarrow 1$
13:     **else**
14:         **if** has_color($n_t$) **then**
15:             $c_t \leftarrow$ color($n_t$)
16:         **else**
17:             $c_t \leftarrow$ color(par($n_t$))
18:         $\delta_{\tau,c_t} = |\{\ell : \lfloor \gamma^\tau \rfloor < \ell \leq t, \; c_\ell = c_t\}|$
19:         $w_t \leftarrow 1/\left(M_{\tau,c_t} + \delta_{\tau,c_t}\right)$
20:     // Incrementally update tree and induced graph
21:     $Q_{t+1} \leftarrow Q_t \backslash \{n_t\} \cup \mathcal{C}(n_t)$
22:     $V_0 \leftarrow V_0 \cup \mathcal{C}(n_t)$
23:     $E_0 \leftarrow E_0 \cup \{(n_t, n') : n' \in \mathcal{C}(n_t)\}$
24:     // Clustering is expensive, we amortize the cost over the search steps using an exponential schedule
25:     **if** $t = r$ **then**
26:         // Cluster Step
27:         $\{G_1, \ldots, G_N\} \leftarrow$ leiden($G_0 = (V_0, E_0)$)
28:         // Update node colors and color count map
29:         $(V_k, E_k) \leftarrow G_k$
30:         **for** $i, U$ **in** enumerate($V_k$) **do**
31:             **for** $v$ **in** $U$ **do**
32:                 color($v$) $\leftarrow i$
33:         $M_{\tau+1,c} \leftarrow |\{v \in V_0 : \text{color}(v) = c\}|$ for all $c \in \{1, \ldots, |V_k|\}$
34:         $r \leftarrow \lceil r * \gamma \rceil$
35:         $\tau \leftarrow \tau + 1$
36:     $t \leftarrow t + 1$
37: **return** False

---

---

**Algorithm 2** $\sqrt{\text{LTS}}$-H

---

1: **Input:** Root node $n_1$, budget $b > 0$, policy $\pi_\theta$, heuristic $h_\omega$, heuristic temperature $\alpha$
2: **Output:** Solution node, or status of failed search
3: $t \leftarrow 1$
4: $Q_t \leftarrow \{n_1\}$
5: **while** $Q_t \neq \{\}$ **and** $t \leq b$ **do**
6:      $n_t = \underset{n \in Q_t}{\arg\min} \; \underset{n_i \prec n}{\min} \; \frac{1}{w_i} \left( \sum_{n_i \prec n' \preceq n} \frac{1}{\pi_\theta(n'|n_i)} \right)$
7:      **if** `is_solution`$(n_t)$ **then**
8:          **return** $n_t$
9:      **if** `is_root`$(n_t)$ **then**
10:      $w_t \leftarrow 1$
11:      **else**
12:      $w_t \leftarrow \exp\left(-\alpha \frac{h_\omega(n_t)}{h_\omega(n_1)}\right)$
13:      $Q_{t+1} \leftarrow Q_t \backslash \{n_t\} \cup \mathcal{C}(n_t)$
14:      $t \leftarrow t + 1$
15: **return** `False`

---

---

**Algorithm 3** $\sqrt{\text{LTS}}$-LH

---

1: **Input:** Root node $n_1$, budget $b > 0$, policy $\pi_\theta$, graph update factor $\gamma > 1$, cluster level $k$, heuristic temperature $\alpha$, mixing coefficients $u_a$ and $u_b$.
2: **Output:** Solution node, or status of failed search
3: $Q_1 \leftarrow \{n_1\}$
4: $V_0 \leftarrow \{n_1\}$, $E_0 \leftarrow \{\}$
5: $t \leftarrow 1, r \leftarrow 1, \tau \leftarrow 1$
6: **while** $Q_t \neq \{\}$ **and** $t \leq b$ **do**
7:    $n_t = \underset{n \in Q_t}{\arg\min} \min_{n_i \prec n} \frac{1}{w_i} \left( \sum_{n_i \prec n' \preceq n} \frac{1}{\pi_\theta(n'|n_i)} \right)$
8:    **if** is_solution($n_t$) **then**
9:      **return** $n_t$
10:    // Find node's color
11:    **if** is_root($n_t$) **then**
12:      $w_t \leftarrow 1$
13:    **else**
14:      **if** has_color($n_t$) **then**
15:        $c_t \leftarrow$ color($n_t$)
16:      **else**
17:        $c_t \leftarrow$ color(par($n_t$))
18:      $\delta_{\tau,c_t} = |\{\ell : \lfloor \gamma^\tau \rfloor < \ell \leq t, \ c_\ell = c_t\}|$
19:      $w_t \leftarrow u_a \left( M_{\tau,c_t} + \delta_{\tau,c_t} \right)^{-1} + u_b \exp\left( -\alpha \frac{h_\omega(n_t)}{h_\omega(n_1)} \right)$
20:    // Incrementally update tree and induced graph
21:    $Q_{t+1} \leftarrow Q_t \backslash \{n_t\} \cup \mathcal{C}(n_t)$
22:    $V_0 \leftarrow V_0 \cup \mathcal{C}(n_t)$
23:    $E_0 \leftarrow E_0 \cup \{(n_t, n') : n' \in \mathcal{C}(n_t)\}$
24:    // Clustering is expensive, we amortize the cost over the search steps using an exponential schedule
25:    **if** $t = r$ **then**
26:      // Cluster Step
27:      $\{G_1, \ldots, G_N\} \leftarrow$ leiden($G_0 = (V_0, E_0)$)
28:      // Update node colors and color count map
29:      $(V_k, E_k) \leftarrow G_k$
30:      **for** $i, U$ **in** enumerate($V_k$) **do**
31:        **for** $v$ **in** $U$ **do**
32:          color($v$) $\leftarrow i$
33:      $M_{\tau+1,c} \leftarrow |\{v \in V_0 : \text{color}(v) = c\}|$ for all $c \in \{1, \ldots, |V_k|\}$
34:      $r \leftarrow \lceil r * \gamma \rceil$
35:      $\tau \leftarrow \tau + 1$
36:    $t \leftarrow t + 1$
37: **return** False

---

# B. Proof of the Theorem 3.1

In every search iteration, the following operations occur: the min-cost node is removed from the priority queue, the rerooting weight is computed, and the children nodes are added back into the priority queue.

Extracting the min-cost node from a priority queue with $M$ items is $O(\log M)$. Computing the rerooting weight for the expanded node is $O(d(n))$ with depth $d(n)$. If $b$ is the branching factor, then inserting the $b$ children into the priority queue is $O(b \log M)$. The cost of each node is computed during insertion to the priority queue. Equation 2 requires a traversal of all ancestor nodes. We note that Orseau et al. (2024) show how this can be reduced to an $O(1)$ computation for most cases, but for the sake of this analysis we assume all ancestor node information is required. Over $N$ steps, the runtime for these computations is given by

$$O\left(\sum_{i=1}^{N}(1+b)\log M_i + d_i\right), \tag{9}$$

where $M_i$ is the size of the priority queue at step $i$ and $d_i$ is the depth of node $n_i$. Since $M_i \leq bN$ and $d_i \leq D$, this simplifies to $O(bN \log N + ND)$.

The frequency with which a Leiden clustering is performed follows a geometric series with common ratio $\gamma$. With a budget $N$, there are $\lfloor \log_\gamma(N) \rfloor$ calls to `leiden`. The Leiden algorithm has a time complexity of $O(L|E|)$, where $L$ is the number of hierarchical cluster graphs created and $|E|$ is the number of edges in the input graph (Sahu et al., 2024). Since the cluster level $L = k$ is a constant, and the size of the edge set of the input graph is bounded by a constant factor of the size of the vertex set, each call to `leiden` has complexity $O(b\gamma^k) = O(\gamma^k)$, where $\gamma^k$ is the iteration `leiden` is called on a graph of size $b\gamma^k$. Over the $N$ steps, the runtime for the calls to `leiden` is given by

$$O\left(\sum_{i=0}^{\lfloor \log_\gamma N \rfloor} b\gamma^i\right) = O\left(\frac{b\left(\gamma^{1+\log_\gamma N} - 1\right)}{\gamma - 1}\right) = O\left(\frac{bN\gamma}{\gamma - 1}\right), \tag{10}$$

Which is linear so long as $\gamma > 1 + 1/\epsilon$ where $\epsilon > 0$ is a constant. Therefore, the overall runtime of $\sqrt{\text{LTS}}$-L is $O(bN \log N + ND)$ as the additional cost of the Leiden algorithm has an amortized constant cost.

# C. Implementation and Machine Details

The algorithms and environment are implemented in C++, adhering to the C++23 standard. The code[1] is compiled using the *GNU Compiler Collection* version 15.2.0, and uses the PyTorch 2.7 C++ frontend (Paszke et al., 2019). The Leiden clustering subroutine uses `leidenalg`[2], an efficient open source implementation. Where available, the official implementation of comparison methods are used. All experiments used 8 threads for the bootstrap training and testing, and were conducted on an Intel i9-7960X and Nvidia 3090, with 128GB of system memory running Ubuntu 24.04.

# D. Additional Experimental Details

During the online bootstrap training process, all algorithms start with an initial budget of 4,000 node expansions. Batched inference is used during training, where generated nodes are placed into a queue before sending to the policy/heuristic networks for inference. A batch size of 32 is used, with smaller batches being used if the open priority queue becomes empty before 32 additional nodes are generated. In our experiments, all algorithms use ResNet-based (He et al., 2016) networks. Algorithms which use both a policy and heuristic use a single network with two heads. The networks for WA*, $\sqrt{\text{LTS}}$-L, $\sqrt{\text{LTS}}$-H and $\sqrt{\text{LTS}}$-LH use 8 blocks of 128 ResNet channels, are trained using the Adam optimizer (Kingma, 2014), with learning rate of 3E-4 and L2-regularization of 1E-4. The baselines LTS($\pi^{\text{SG}}$) and PHS*($\pi^{\text{SG}}$) are trained following Tuero et al. (2025) and use the open source implementation.[3]

For $\sqrt{\text{LTS}}$-L and $\sqrt{\text{LTS}}$-LH, we use a graph update factor of $\gamma = 1.2$ for the geometric schedule for when the Leiden clustering procedure is run. We use a cluster level $k = N$. For $\sqrt{\text{LTS}}$-H and $\sqrt{\text{LTS}}$-LH, we use $\alpha = 10$ for the inverse-temperature heuristic parameter. For an ablation on the impact of the value of $\alpha$ used, see Appendix H. All reported results use $u_a = u_b = 1$ for $\sqrt{\text{LTS}}$-LH. For a sensitivity analysis on having balanced rerooters, see Appendix G.

---

[1]https://github.com/tuero/siirlts2026

[2]https://github.com/vtraag/libleidenalg

[3]https://github.com/tuero/subgoal-guided-policy-search

## E. Ablation: Impact of Clustering Level

The Leiden clustering algorithm takes a base graph $G_0$ and produces a hierarchy of cluster graph $\{G_1, \ldots, G_N\}$. The selected graph at level $k \in [1, N]$ influences what structures are used when coloring nodes, which the rerooters $\sqrt{\text{LTS}}$-L and $\sqrt{\text{LTS}}$-LH use to compute their rerooting weights.

*Table 3.* Training results of $\sqrt{\text{LTS}}$-L over the BoulderDash 20% environments of varying cluster levels.

| Cluster Level ($K$) | Expansions | Time (Hours) |
|:---:|:---:|:---:|
| 1 | 166,690,697 | 42.86 |
| $N/2$ | **65,307,017** | **20.38** |
| $N$ | 77,504,769 | 21.44 |

Table 3 shows the bootstrap training loss for $\sqrt{\text{LTS}}$-L over the BoulderDash 20% dirt environment of varying cluster levels. A cluster level of $k = N/2$ which corresponds to half the number of cluster graphs completes training in the fewest number of expansions and time, but using $k = N$ is also competitive. We note that this will be dependent on the implementation of the Leiden algorithm being used as different approximations and optimizations can be used. See Appendix C for details and the specific implementation used.

## F. Ablation: Impact of Graph Update Frequency

The cluster graphs from the Leiden algorithm are built using an update schedule which follows a geometric growth strategy with factor $\gamma$. How frequently the graphs are updated will have two major consequences. One is the overall speed of the algorithm, where using a smaller $\gamma$ will result in more graphs being built under a fixed budget. Creating cluster graphs more frequently means that more time is spent in the `leiden` procedure than actually running the search, resulting in a lower number of nodes-per-second which can be achieved. The second is that using a larger $\gamma$ means that there will be more expansions between graph updates, with a larger percentage of those nodes being generated after the most recent cluster graph. If a node is expanded which does not have a color assigned to it (as is the case when an expanded node was generated after the most recent graph clustering), then an approximation is used where we assume that node gets the same color as its parent. A larger $\gamma$ means that a larger percentage of the nodes will have an approximated color.

*Table 4.* Test results of $\sqrt{\text{LTS}}$-L on the Sokoban environment of varying cluster graph update factors.

| Cluster Update Frequency ($\gamma$) | Expansions | Time (Seconds) | Nodes per Second |
|:---:|:---:|:---:|:---:|
| 1.05 | 2,932.19 | 2.01 | 7.57 |
| 1.10 | 2,664.08 | 1.36 | 8.00 |
| 1.20 | 2,715.87 | 1.42 | 8.47 |
| 1.50 | 2,835.79 | 1.49 | 8.71 |
| 2.00 | 3,187.88 | 1.71 | 8.67 |
| 4.00 | 3,357.53 | 2.12 | 9.24 |

Table 4 shows the test results of using the same trained policy on the Sokoban environment, but with varying values of $\gamma$. In general, as $\gamma$ increases, the nodes per second that the algorithm can achieve increases. We also see that as $\gamma$ decreases, the number of expansions decreases, as expected, because because the rerooter can make greater use of the underlying state-space structure to assign informative weights rather than relying on approximations.

## G. Ablation: Balanced Rerooters

The constant $C \geq 1$ in Theorem 3.2 signals that the bound is related to how *balanced* the two sub-rerooter components are. Some natural questions to ask are how balanced are the two sub-rerooters in $\sqrt{\text{LTS}}$-LH, and how sensitive are the observed node expansions to the factor $C$ in Theorem 3.2.

To answer these questions, we ran an additional sensitivity study in Sokoban, fixing $u_b = 1$ and varying $u_a$, which are the *weighting* applied to each sub-rerooter component. These results are summarized in Table 5. Performance varies only modestly across a broad range of ratios, suggesting that the hybrid rerooter is not highly sensitive to exact tuning. The best

results occur when the two signals are roughly balanced, *i.e.*, when $w_{a,<T}/w_{b,<T} \approx 1$. In practice, $u_a = u_b = 1$ is a good default. One can then inspect the observed ratio $w_{a,<T}/w_{b,<T}$ and rescale if needed, but fine-tuning is not critical.

*Table 5.* Sensitivity analysis of unbalanced rerooters. $\sqrt{\text{LTS}}$-LH is tested on the Sokoban environment, with varying ratios of $u_a$ to $u_b$. Time is measured in seconds.

| $(u_a, u_b)$ | EXPANSIONS | SOLUTION COST | TIME (S) | $w_{a,<T}/w_{b,<T}$ |
|---|---|---|---|---|
| (1.00, 1.00) | 1,736.03 | 36.37 | 1.10 | 2.03 |
| (0.75, 1.00) | 1,477.30 | 36.14 | 0.90 | 1.31 |
| **(0.60, 1.00)** | **1,400.88** | 36.05 | 0.84 | **0.99** |
| (0.50, 1.00) | 1,523.21 | 36.00 | 0.92 | 0.81 |
| (0.25, 1.00) | 1,509.74 | 35.89 | 0.90 | 0.44 |

## H. Ablation: Heuristic Rerooter Temperature

The parameter $\alpha$ in $\sqrt{\text{LTS}}$-H and $\sqrt{\text{LTS}}$-LH acts as an inverse-temperature parameter. When $\alpha$ is small, the weights are more similar across nodes, so the rerooter behaves conservatively. As $\alpha$ increases, the weight mass becomes more concentrated on nodes whose heuristic values are small relative to the root, causing rerooting to focus more aggressively on regions that appear to have made substantial progress toward a goal.

Table 6 shows the result of varying the $\alpha$ inverse-temperature parameter on the Sokoban environment for the $\sqrt{\text{LTS}}$-LH algorithm. The same trained policy and heuristic networks were used for all test runs, with $(u_a, u_b) = (0.60, 1.00)$ as this level of mixing resulted in balanced rerooters, as shown in Appendix G. The value of $\alpha = 10$ solves the Sokoban test problems in the fewest number of expansions, which is the value of $\alpha$ used in all other experiments. As $\alpha$ decreases towards 1, the number of expansions increases due to the rerooting weight contribution from the heuristic rerooter being similar across all nodes. Larger values of $\alpha$ allows the search to focus on areas of the tree where progress is being made towards the goal. However, if $\alpha$ becomes too large, then the search can overcommit to what seems like progress being made but what could be a deadlocked subtree.

*Table 6.* Sensitivity analysis of heuristic inverse-temperature parameter. $\sqrt{\text{LTS}}$-LH is tested on the Sokoban environment, with varying values of $\alpha$. Time is measured in seconds.

| $\alpha$ | EXPANSIONS | LENGTH | TIME (S) |
|---|---|---|---|
| 1 | 2,949.79 | 37.45 | 2.36 |
| 5 | 1,959.70 | 36.20 | 1.25 |
| 10 | **1,400.88** | 36.05 | 0.84 |
| 20 | 1,699.64 | 36.47 | 1.13 |

## I. Ablation: Analysis of Clustering-based Rerooters

From the results in Table 1 and Table 2, $\sqrt{\text{LTS}}$-L can perform quite a bit worse than $\sqrt{\text{LTS}}$-H and $\sqrt{\text{LTS}}$-LH. The usefulness of the information provided by the clusters found which $\sqrt{\text{LTS}}$-L relies on depends on the underlying structure of the state-space. The clustering rerooter is most effective when the induced state graph has well-separated regions connected by narrow bottlenecks, because entering a new cluster then signals genuine progress. When comparing against PHS*($\pi^{\text{SG}}$) in Table 1, $\sqrt{\text{LTS}}$-L performs relatively worse in the Sokoban and TSP GridWorld environments. Since $\sqrt{\text{LTS}}$-L reached the time limit during training on the CraftWorld environment and did not fully complete the bootstrap process, we omit from this analysis. In Table 2, as the complexity of the BoulderDash environments increases, $\sqrt{\text{LTS}}$-L continues to degrade relative to the other rerooters.

To quantify this, we measured the average number of edges crossing each cluster boundary in the induced state graph, averaged over problems. Lower values indicate sharper separation. These results are given in Table 7. Under this metric, BoulderDash exhibits a clear progression as the domain becomes less structured, CraftWorld remains relatively low, Sokoban and TSP are substantially higher. In Sokoban, this helps explain why the clustering rerooter is less effective than the strongest baselines, and in TSP, the very large value is consistent with the strongest degradation, where the clustering rerooter requires

*Table 7.* Average number of edges crossing different clusters. A lower value indicates sharper separation between clusters.

| ENVIRONMENT | AVG EDGE CROSSING PER CLUSTER |
|---|---|
| BOULDERDASH (10%) | 7.6219 |
| BOULDERDASH (20%) | 8.8399 |
| BOULDERDASH (30%) | 11.0756 |
| BOULDERDASH (40%) | 13.6613 |
| BOULDERDASH (50%) | 16.6101 |
| CRAFTWORLD | 9.7805 |
| SOKOBAN | 16.6344 |
| TSP (GRIDWORLD) | 35.2884 |

nearly twice as many expansions as the heuristic/hybrid rerooters (despite both test-time results being overall low). Overall, these results support our claim that the usefulness of clustering-based rerooting depends strongly on the presence of sharp, bottleneck-like state-space structure, and that this metric provides a quantitative way to characterize when that structure is present or absent.

## J. Additive Rerooters

In this section we demonstrate how additive rerooters can take advantage not just of the best of the sub-rerooters, but more importantly of the *synergy* of the sub-rerooters. Let us start with an example, before giving the proof of Theorem 3.2.

**Example J.1** (Two rerooters). For example, suppose $d(n^*) = 30$ in a perfect binary tree with a uniform policy. Then the LTS bound tells us that LTS would take at most $(d(n^*) + 1)2^{30}$ node visits to find $n^*$, but the refined algorithm and bound using $c_1^r$ actually takes at most $c_1^r(n^*) = 1 + 2 + 4 + \ldots 2^{30} = 2^{31} - 1$ node visits (Orseau et al., 2024). Let $n^a \prec n^*$ such that $d(n^a) = 10$. Let $n^b \prec n^*$ such that $d(n^b) = 20$.

Let us consider three rerooters $w_a, w_b, w_{ab}$. $\sqrt{\text{LTS}}$ with $w_a$ visits $n^a$ at step $T_{a,a}$ and $n^*$ at step $T_a^*$. $\sqrt{\text{LTS}}$ with $w_b$ visits $n^b$ at step $T_{b,b}$ and $n^*$ at step $T_b^*$. $\sqrt{\text{LTS}}$ with $w_{ab}$ visits $n^a$ at step $T_{ab,a}$ and $n^b$ at step $T_{ab,b}$ and $n^*$ at step $T_{ab}^*$. We set $w_{a,1} = w_{b,1} = w_{a,T_{a,a}} = w_{b,T_{b,b}}$, and 0 everywhere else. Furthermore, define $w_{ab} = w_a + w_b$.

Equation (4) tells us that for $\sqrt{\text{LTS}}$ with rerooter $w_a$ is competitive with the best subtask decomposition. Therefore, it is also competitive with any subtask decomposition that is meaningful for analysis. Let us compare it with the subtask decomposition $n_1 \prec n^a = n_{T_{a,a}} \prec n^*$ (which happens to be the best one, in this case), the node $n^*$ is visited at $T_a^*$ with

$$T_a^* - 1 \leq w_{a,<T_a^*} \max \left\{ \frac{c_1^r(n^a)}{w_{a,1}}, \frac{c_{T_{a,a}}^r(n^*)}{w_{a,a}} \right\} = 2 \max \left\{ \frac{2^{11} - 2}{1}, \frac{2^{21} - 2}{1} \right\} = 2^{22} - 4 \,.$$

This is already an improvement over the $2^{31}$ node visits of LTS. Similarly, for $\sqrt{\text{LTS}}$ with rerooter $w_b$, considering the subtask decomposition $n_1 \prec n^b = n_{T_{b,b}} \prec n^*$, the node $n^*$ is visited at step $T_b^*$ with

$$T_b^* - 1 \leq w_{b,<T_b^*} \max \left\{ \frac{c_1^r(n^b)}{w_{b,1}}, \frac{c_{T_{b,b}}^r(n^*)}{w_{b,b}} \right\} = 2 \max \left\{ \frac{2^{21} - 2}{1}, \frac{2^{11} - 2}{1} \right\} = 2^{22} - 4 \,.$$

But for $\sqrt{\text{LTS}}$ with rerooter $w_{ab} = w_a + w_b$, we can consider the subtask decomposition $n_1 \prec n^a \prec n^b \prec n^*$ since both $n^a$ and $n^b$ now have non-zero rerooting weights:

$$T_{ab}^* - 1 \leq w_{ab,<T_{ab}} \max \left\{ \frac{c_1^r(n^a)}{w_{ab,1}}, \frac{c_{T_{ab,a}}^r(n^b)}{w_{ab,a}}, \frac{c_{T_{ab,b}}^r(n^*)}{w_{ab,b}} \right\} = 4 \max \left\{ \frac{2^{11} - 2}{1}, \frac{2^{11} - 2}{1}, \frac{2^{11} - 2}{1} \right\} = 2^{13} - 8$$

$$\approx 4\sqrt{T_a^*} \,.$$

Hence, the hybrid rerooter $w_{ab}$ benefits from the synergy of $w_a$ and $w_b$ together, by decomposing into 3 subtasks rather than two, leading to a $512\times$ speedup. $\qquad \square$

We propose a more general result, as a variant of the subtask decomposition bound of Equation (4) that takes advantage of the synergy of *balanced* additive rerooters.

**Theorem J.2** (Theorem 3.2, Balanced additive rerooters). *Let $w_a$ and $w_b$ be two rerooters, with relative weights $u_a \geq 0$ and $u_b \geq 0$. Let $w = u_a w_a + u_b w_b$ be the rerooter used by $\sqrt{\text{LTS}}$. When visiting the node $n_T$ at step $T$, for any $C \geq 1$ satisfying $\frac{1}{C} \leq \frac{u_a w_{a,<T}}{u_b w_{b,<T}} \leq C$, then the number of node visits is bounded by*

$$T \leq 1 + (C+1) \times \min_{D \in \mathcal{D}(n^*)} \max_{i < |D|} \min\left\{\frac{w_{a,<T}}{w_{a,T_i}}, \frac{w_{b,<T}}{w_{b,T_i}}\right\} c^r_{T_i}(n_{T_{i+1}}).$$

*Proof.* Using the assumption on the rerooters and assuming $C$ exists (i.e., all quantities are strictly positive), $u_a w_{a,<T} \leq C u_b w_{b,<T}$ and $u_b w_{b,<T} \leq C u_a w_{a,<T}$, so $w_{<T} = u_a w_{a,<T} + u_b w_{b,<T} \leq (C+1)u_a w_{a,<T}$ and also $w_{<T} \leq (C+1)u_b w_{b,<T}$. Therefore,

$$\frac{w_{<T}}{w_t} = \frac{w_{<T}}{u_a w_{a,t} + u_b w_{b,t}}$$

$$\leq \min\left\{\frac{w_{<T}}{u_a w_{a,t}}, \frac{w_{<T}}{u_b w_{b,t}}\right\} \leq (C+1)\min\left\{\frac{u_a w_{a,<T}}{u_a w_{a,t}}, \frac{u_b w_{b,<T}}{u_b w_{b,t}}\right\} = (C+1)\min\left\{\frac{w_{a,<T}}{w_{a,t}}, \frac{w_{b,<T}}{w_{b,t}}\right\}.$$

Plugging in to Equation (4) gives the result. $\square$

As mentioned in the main text, the constant $C \geq 1$ can be controlled by adjusting the weights $u_a$ and $u_b$, such that $u_a w_{a,<T}$ and $u_b w_{b,<T}$ are close to one another.

# K. Complete Training Results

*Table 8.* Average online training loss with respect to expansions and time. Time measures the sum an algorithm spends on each problem across all threads used during training, in hours.

| ALGORITHM | EXPANSIONS | TIME | SOLVED |
|---|---|---|---|
| **BOULDERDASH** | | | |
| WA*(1.5) | 652,027,388.80 | 122.87 | 9,966.60 |
| LTS | 1,462,860,716.80 | 278.12 | 106.00 |
| LTS($\pi^{SG}$) | 259,846,338.00 | 104.19 | 9,999.60 |
| PHS*($\pi^{SG}$) | 94,266,331.60 | 35.12 | 10,000.00 |
| $\sqrt{\text{LTS}}$-L | 128,970,198.60 | 32.57 | 9,996.00 |
| $\sqrt{\text{LTS}}$-H | **24,154,363.00** | **7.46** | 9,999.40 |
| $\sqrt{\text{LTS}}$-LH | 28,257,820.80 | 8.47 | 9,995.80 |
| **CRAFTWORLD** | | | |
| WA*(1.5) | 456,670,096.60 | 155.71 | 9,847.40 |
| LTS | 804,101,431.20 | 277.91 | 315.00 |
| LTS($\pi^{SG}$) | 422,689,637.00 | 278.07 | 434.20 |
| PHS*($\pi^{SG}$) | 252,454,918.60 | 153.53 | 9,998.20 |
| $\sqrt{\text{LTS}}$-L | 479,154,851.00 | 257.24 | 9,507.40 |
| $\sqrt{\text{LTS}}$-H | 164,845,073.20 | 60.20 | 9,939.40 |
| $\sqrt{\text{LTS}}$-LH | **155,766,283.20** | **48.51** | 9,925.00 |
| **SOKOBAN** | | | |
| WA*(1.5) | 131,496,170.40 | 18.77 | 47,411.80 |
| LTS | 172,870,658.20 | 29.86 | 47,378.00 |
| LTS($\pi^{SG}$) | 233,050,490.80 | 69.79 | 47,226.20 |
| PHS*($\pi^{SG}$) | 127,096,718.60 | 38.15 | 47,452.00 |
| $\sqrt{\text{LTS}}$-L | 170,776,344.60 | 34.91 | 47,564.20 |
| $\sqrt{\text{LTS}}$-H | 100,488,660.60 | 17.89 | 47,798.60 |
| $\sqrt{\text{LTS}}$-LH | **95,196,226.80** | **17.88** | 47,756.60 |
| **TSP (GRIDWORLD)** | | | |
| WA*(1.5) | 2,059,512,675.20 | 258.51 | 4,000.60 |
| LTS | 1,654,329,587.60 | 226.23 | 9,999.60 |
| LTS($\pi^{SG}$) | 122,659,044.00 | 41.26 | 10,000.00 |
| PHS*($\pi^{SG}$) | 22,764,606.80 | 7.21 | 10,000.00 |
| $\sqrt{\text{LTS}}$-L | 67,925,838.20 | 13.91 | 10,000.00 |
| $\sqrt{\text{LTS}}$-H | 693,847,701.00 | 53.30 | 10,000.00 |
| $\sqrt{\text{LTS}}$-LH | **13,481,743.40** | **3.33** | 10,000.00 |

