# OpenReview forum: "Structure-Induced Information for Rerooting Levin Tree Search"
_ICML.cc/2026/Conference — ICML 2026 regular_

### Official Review · Reviewer_9T9e · 2026-03-11

**Soundness:** 2
**Presentation:** 3
**Significance:** 3
**Originality:** 3
**Overall Recommendation:** 4
**Confidence:** 3

**Summary:**

### Motivation

Levin Tree Search (LTS) is a general‑purpose, anytime planner that interleaves breadth‑first search with a probabilistic bias derived from a policy.

Previously,, LTS uses a *rerooting weight* that is essentially uniform, which can be sub‑optimal when the search space has strong geometric structure.

### Proposal

The authors propose two weighting mechanisms to reduce the number of loop iterations (or expansions) required to find a solution: a clustering-based rerooter that captures the global structure of the state space, and a heuristic-based rerooter that leverages local cost-to-go information.

**Global Structure Inducing weights (H)**

First, the Leiden clustering algorithm is used to create clusters of states at increasing levels of granularity (hierarchy).

Then, using a specified hierarchy level $k$ (a hyperparameter), each state is colored with its cluster index $t$.

Then the weights are can be computed as: $w_t = \frac{1}{M_{\tau,c_t} + \delta_{\tau,c_t}}$, where $M_{\tau,c_t} + \delta_{\tau,c_t}$ is an estimate of the cluster size.

**Local Structure–Induced Rerooting (L)**

The local weights are computed as: $w_t = \texttt{clamp}(0, 1-\frac{h(n_t)}{h(n_1)}, 1)$, where $h(\cdot)$ is a heuristic estimate.

**Hybrid Structure–Induced Rerooting (LH)**

The authors combine these two approaches by adding the weights to yield the hybrid weight.

Theoretical analysis shows that:

- The runtime of $\sqrt{}$LTS‑L matches breadth‑first search up to a constant factor (the algorithm never expands more nodes than a standard BFS of the same depth): $O(bN \log N + DN)$.
- The additive combination of the two sub‑rerooters yields a bound that is at least as good as the best of the two individual bounds.

### Experimental results

The method is tested in environments: BoulderDash, CraftWorld, Sokoban, and Travelling Salesman.

Baselines: Weighted A*, LTS, LTS($\pi^\text{SG}$), PHS*($\pi^\text{SG}$).

- While nearly all baselines solve the tasks, $\sqrt{}$LTS performs optimally. The $\sqrt{}$LTS variants beat the baselines at the number of expansions on 2/4 tasks (Table 1), and 1/4 tasks on the path length.
- On increasing levels of difficulty in the BoulderDash task (10%, 20%,…, 50%), $\sqrt{}$LTS variants perform well (compared only to PHS*) with the hybrid weights (LH) leading in 2/5 levels, H weights leading in 3/5 tasks (Table 2).

**Compliance With Llm Reviewing Policy:**

Affirmed.

**Final Justification:**

I am raising my score to weak accept since the paper addresses an existing knowledge gap. However, I am not entirely satisfied with the discrepancy between the theoretical and empirical results. Though the authors have committed to changing it to 'within a factor C', it is still not entirely true, as the results do not show the performance difference converging to a constant factor C; instead, the gap grows in Table 2 (scaling with difficulty).

If the theoretical results do not reflect in the experiments, they are misleading.

**Key Questions For Authors:**

- Please address the concerns in the Weaknesses.

**Limitations:**

No discussion on limitations.

**Strengths And Weaknesses:**

### Strengths

- Authors present a method to bias or prioritize the search for a solution based on heuristics.
- It is theoretically proven that the heuristics can be combined additively.

### Weaknesses

- The method is presented as a modification to the $\sqrt{}$LTS algorithm. If so, it is crucial to include the $\sqrt{}$LTS as a baseline to accurately measure the performance differences induced by the proposals. Without that baseline, the claim for the performance gain is not justified.
- While the additive combination of weights is shown to be theoretically sound, the results show that the hybrid weights (LH) are not clearly superior. The main results show that hybrid weights outperform H weights only on 2/4 tasks. In Table 2, a clear trend can be seen where H outperforms LH as task difficulty increases (and the difference widens with difficulty). The scaling of the performance difference between H and LH with difficulty raises a question whether the second theoretical claim, “The additive combination yields a bound that is at least as good as the best of the two individual bounds”, is sound.
- No discussion on limitations.

---

> ### Author Rebuttal · Authors · 2026-03-29
>
> Thank you for your helpful comments and feedback.
>
> **Question 1**:
> The only $\sqrt{LTS}$ baseline possible is the one where we use a rerooter which assigns a rerooting weight of 1 for the initial state, and 0 elsewhere.
> There isn't any other version of $\sqrt{LTS}$ that we could use.
> This baseline is equivalent to LTS [1], which we already include in our experiments.
> Our work is providing the first known implementation of a rerooter that is different from the one that represents LTS.
>
> [1] Laurent Orseau, Marcus Hutter, Levi H. S. Lelis: Exponential Speedups by Rerooting Levin Tree Search. CoRR abs/2412.05196 (2024)
>
> **Question 2**:
> The quoted statement is not the claim proved in our paper.
> The theory is not contradicted by Table 2: Theorem 3.2 gives the hybrid rerooter a controlled upper bound, but because that bound includes a multiplicative balance factor $C$, it does not imply that the hybrid must empirically outperform both components on every task or difficulty level.
> Thus, cases where $\sqrt{LTS}\text{-H}$ slightly outperforms $\sqrt{LTS}\text{-LH}$ are still fully consistent with the theory.
> More precisely, Theorem 3.2 analyzes the additive rerooter $w = u_a w_a+ u_b w_b$ under a balanced condition and shows that it inherits a valid expansion bound from its components, while allowing the possibility of strict improvements when the two rerooters provide complementary information on different parts of the decomposition.
> We also agree that one sentence in the experimental discussion was too broad: “The hybrid rerooter is never worse than either component.”
> Since Table 2 does show settings where $\sqrt{LTS}\text{-H}$ slightly outperforms $\sqrt{LTS}\text{-LH}$, we will revise this to the more accurate claim that the hybrid is competitive with its components and can benefit from combining complementary signals.
>
> **Question 3**:
> We would be happy to include a note about the limitation of rerooting methods as a whole.
> More broadly, rerooting methods are most effective when the search space contains informative intermediate structure that can be translated into useful rerooting weights.
> As a result, rerooting is not a free gain in every domain. When the induced structure is weak, unstable, or only loosely aligned with true progress toward the goal, rerooting may mainly redistribute search effort rather than substantially improve it.
> Empirically, this shows up primarily as variation across domains rather than a failure of the framework.
> For example, Sokoban appears to provide less informative clustering structure than the other environments we study.
> At the same time, these limitations also motivate the hybrid rerooter, which combines complementary structural signals and was the most robust variant across the domains we tested. Finally, our experiments focus on single-agent deterministic planning under online bootstrap training, so evaluating rerooting in broader settings remains an important direction for future work.

---

> > ### Author Rebuttal · Reviewer_9T9e · 2026-04-03
> >
> > I thank the authors for the response. It, along with responses to other reviewers, addresses most of my concerns. If the authors revise the claims to better reflect the contributions and add the limitations, I will raise my score. I am slightly concerned that we don't have a baseline rerouter from the original paper (Orseau et al., 2024), but if so, I agree it is a genuine knowledge gap.

---

> > > ### Author Response · Authors · 2026-04-04
> > >
> > > We thank the reviewer for taking the time to read through our rebuttal to them, as well as the rebuttal to the other reviewers, as their feedback helps make the paper stronger.
> > >
> > > We will revise the paper as follows.
> > >
> > > Instead of writing "The hybrid rerooter is never worse than either component.", we will write "The hybrid rerooter is at least as good (within a factor C) ..." This is to avoid confusion between our theoretical and empirical results.
> > >
> > > We will also add the limitations paragraph as explained in our rebuttal.
> > >
> > > As for the rerooter baseline, we would like to emphasize that there is no concrete $\sqrt{\mathrm{LTS}}$ algorithm that we can use from the prior literature.
> > > The paper [1] was strictly theoretical and did not provide any practical means of learning a rerooter. The closest we have is LTS, which we used as a baseline. LTS is a special case of $\sqrt{\mathrm{LTS}}$. We are the first to introduce a method to learn a rerooter.
> > >
> > > **Important:** ICML does not allow us to update our paper at this point; we will do so in an eventual camera-ready revision. Given that we have clarified the reviewer's questions, we kindly ask them to update their score to avoid sending the wrong message to the AC handling our paper. We thank them in advance for doing so.
> > >
> > > [1] Laurent Orseau, Marcus Hutter, Levi H. S. Lelis: Exponential Speedups by Rerooting Levin Tree Search. CoRR abs/2412.05196 (2024)

---

### Official Review · Reviewer_y2y8 · 2026-03-12

**Soundness:** 3
**Presentation:** 3
**Significance:** 2
**Originality:** 3
**Overall Recommendation:** 4
**Confidence:** 2

**Summary:**

This paper introduces a new way to improve root-LTS for single-agent deterministic planning problems. Previous root-LTS methods determined rerooter weights through manual design, which lacks flexibility and adds extra burden. The authors propose an automated way to select rerooter weights. They design three weight methods using the global structure of the state space and local heuristics. The authors tested the method on several tasks and the experimental results show that this approach achieves fast training efficiency.

**Compliance With Llm Reviewing Policy:**

Affirmed.

**Final Justification:**

The rebuttal has satisfactorily addressed the majority of my initial concerns regarding the technical details. While the paper appears sound, I remain uncertain about the overall significance and broader impact of the proposed method within the field. Weighing the resolved technical issues against my uncertainty regarding its significance, I have decided to raise my evaluation to a Weak Accept, maintaining a confidence score of 2 to reflect my lingering hesitation on its overall impact.

**Key Questions For Authors:**

key questions:
The proposed algorithm expands more nodes than the baseline models in the Sokoban environment. The paper attributes this to a weaker state-space clustering structure. What specific topological features of Sokoban lead to this suboptimal clustering? Could the authors provide a quantitative analysis of this structural weakness?
The proposed method focuses primarily on discrete and deterministic planning problems. However, many real-world tasks involve continuous state spaces. Could the rerooting mechanism be adapted or applied to continuous control domains?
In the theoretical analysis of the hybrid rerooter, relative weights for the two signals are involved. Did the authors empirically test different weight ratios between the global and local signals? How sensitive is the algorithm's overall search performance to this weight balance?
The current empirical evaluation relies exclusively on Bootstrap training. It is effective for the chosen deterministic environments. However, this raises a question about the framework's methodological generality. Could the proposed structure-induced rerooting mechanism be integrated with standard online reinforcement learning algorithms? There is growing interest in combining tree search with reinforcement learning, discussing or validating the compatibility of root-LTS with online reinforcement learning paradigms would significantly strengthen the paper's broader impact.

**Limitations:**

limitations:
The proposed method focuses primarily on discrete and deterministic planning problems, which restricts its overall applicability. It remains unclear how this framework could generalize to continuous state spaces or environments with stochastic dynamics.

**Strengths And Weaknesses:**

Strengths
1. The technical derivation is rigorous. The mathematical proof is clear. It successfully proves the node expansion limit for the additive rerooter. The experimental design is solid. The method shows higher training efficiency than baselines like SGPS in several complex environments.
2. The core logic is easy to understand. The writing is clear and fluent.
3. The paper solves a key computational challenge in $\sqrt{LTS}$. It successfully bypasses explicit subgoal generation. It uses an adaptive weight selection method instead. This improves the scalability and generalization of the tree search method.
4. The authors combine the existing Leiden clustering algorithm and local heuristic cost estimates. This combination automatically learns the best weights for the rerooter. It offers a fresh algorithmic perspective in the tree search field.
Weaknesses
1. The algorithm does not perform as well as some baselines in the Sokoban and TSP environments. The authors honestly report this fact. This shows the algorithm might have limits under certain topological structures.
2. The method only works for single-agent, discrete, and deterministic problem settings. It cannot move directly to tasks with continuous state spaces or random dynamics.

---

> ### Author Rebuttal · Authors · 2026-03-29
>
> Thank you for your helpful comments and feedback.
>
> **Question 1**
> We agree this deserves a quantitative explanation.
> The clustering rerooter is most effective when the induced state graph has well-separated regions connected by narrow bottlenecks, because entering a new cluster then signals genuine progress.
> Sokoban appears to have a weaker bottleneck structure: many agent/box configurations remain connected through broader cluster boundaries, so clusters are less informative for rerooting.
>
> To quantify this, we measured the average number of edges crossing each cluster boundary in the induced state graph, averaged over problems. Lower values indicate sharper separation.
>
> | Env | Avg # Edge Crossings per Cluster |
> |---|---|
> | BoulderDash (10%) | 7.6219 |
> | BoulderDash (20%) | 8.8399 |
> | BoulderDash (30%) | 11.0756 |
> | BoulderDash (40%) | 13.6613 |
> | BoulderDash (50%) | 16.6101 |
> | CraftWorld | 9.7805 |
> | Sokoban | 16.6344 |
> | TSP Gridworld | 35.2884|
>
> Under this metric, BoulderDash exhibits a clear progression as the domain becomes less structured, CraftWorld remains relatively low, Sokoban and TSP are substantially higher.
> In Sokoban, this helps explain why the clustering rerooter is less effective than the strongest baselines, and in TSP, the very large value is consistent with the strongest degradation, where the clustering rerooter requires nearly twice as many expansions as LTS and the heuristic/hybrid rerooters (despite both test-time results being overall low).
> Overall, these results support our claim that the usefulness of clustering-based rerooting depends strongly on the presence of sharp, bottleneck-like state-space structure, and that this metric provides a quantitative way to characterize when that structure is present or absent.
> We will provide these results into the appendix of our revised paper.
>
>
> **Question 2**:
> Deterministic problems are an important and active area of research.
> Prior work has also studied how transformations from classical planning can be extended to non-classical settings [1], including problems with nondeterminism or incomplete information, beyond the classical planning setting we consider here.
> Even so, we would like to emphasize that tree/graph search algorithms, which are designed for discrete and deterministic environments, are far from niche, and have a long history of being applied to a large variety of domains, including:
> * Combinatorial optimization and Reinforcement Learning
> * Automated Theorem Proving, Program Synthesis, SAT solving
> * Path planning, job shop scheduling, Logistic and GPS routing
> * Computational Game Theory (Go, Chess, etc.)
> * LLM generation "thinking modes" with test-time search (Beam Search, A*, MCTS, LTS, etc.)
>
> However, the reviewers raise a valid point regarding generalizing the approach to stochastic and continuous environments.
> Such extensions represent valuable directions for future work, and we are specifically considering the following methodologies:
> * Extension to continuous dynamics via hierarchical discretization of the state and action spaces.
> * Extension to stochastic environments via determinization (e.g., fixing the random seed during an individual tree search rollout and aggregating results over multiple sampled seeds, assuming access to a forward model or simulator).
>
> [1] Geffner, Hector. "Non-classical planning with a classical planner: The power of transformations." 2014.
>
> **Question 3**:
> Following the reviewer's suggestion, we ran an additional sensitivity study in Sokoban, fixing $u_a=1$ and varying $u_b$.
> Performance varies only modestly across a broad range of ratios, suggesting that the hybrid rerooter is not highly sensitive to exact tuning.
> The best results occur when the two signals are roughly balanced, i.e., when $(u_a w_{a,<T}) / (u_b w_{b,<T}) \approx 1$.
> In practice, $u_a = u_b = 1$ is a good default.
> One can then inspect the observed $w_{a,<T} / w_{b,<T}$ ratio and rescale if needed, but fine-tuning is not critical.
> We will include the following table in the revision.
>
> |($u_a$, $u_b$) | Expansions | Solution Cost | Time (s) | $w_{a,<T} / w_{b,<T}$ |
> |---|---|---|---|---|
> |(1.0, 1.0)| 2,349.73| 36.32| 1.34 | 0.628|
> |(1.0, 0.75)| 2,078.07| 36.25 | 1.13 | 0.765|
> |(1.0, 0.5) | 1,845.22| 36.65| 1.07 | 1.024|
> |(1.0, 0.25) | 2,060.48| 36.91 | 1.24 | 1.514|
>
> **Question 4**:
> Thank you for the suggestion!
> Yes, best-first search algorithms, which include $\sqrt{LTS}$, LevinTS, and A*, can be integrated with RL, provided that the agent has access to a model of the environment (either given or learned).
> The cost function used in best-first search methods determines which node is expanded next, which means the search may *jump* to different parts of the state-space when the search alternates between different branches in the tree.
> This behaviour is only possible with model access, since the agent must evaluate and expand states that are not on the current experienced trajectory.

---

> > ### Author Rebuttal · Reviewer_y2y8 · 2026-04-03
> >
> > Thank the authors for the response. My questions are fully addressed. I will raise my score.

---

### Official Review · Reviewer_wpxn · 2026-03-13

**Soundness:** 3
**Presentation:** 3
**Significance:** 3
**Originality:** 3
**Overall Recommendation:** 4
**Confidence:** 3

**Summary:**

This paper studies rerooting as a way to inject structure into policy-guided tree search without explicit subgoal generation. The work proposes three rerooters for $\sqrt{\mathrm{LTS}}$: a clustering-based rerooter using Leiden clustering over the explored search graph, a heuristic-based rerooter using normalized cost-to-go estimates, and a hybrid additive rerooter combining both signals. In addition to these algorithmic proposals, the paper proves a guarantee for additive rerooters that supports the benefit of combining multiple rerooting signals. The empirical study compares against LTS, SGPS-style baselines, and WA* on several domains including BoulderDash, CraftWorld, Sokoban, and TSP (GridWorld), and reports gains in online training efficiency, especially in harder settings where explicit subgoal methods struggle. The paper positions rerooting as a scalable substitute for explicit subgoal generation and asks a pertinent question about how search effort should be distributed across implicit subtasks in complex planning problems.

**Compliance With Llm Reviewing Policy:**

Affirmed.

**Key Questions For Authors:**

1. How sensitive is the method to the clustering schedule and related hyperparameters? A sensitivity analysis would clarify whether the clustering based rerooter and the hybrid are robust or whether performance depends on careful tuning.

2. The evaluation emphasizes online training curves, but the test time comparison remains incomplete. Under a matched expansion or wall clock budget, how do the resulting policies compare, and how much of the gain remains after accounting for the overhead of Leiden clustering and rerooting bookkeeping on each domain?

3. Since the heuristic rerooter may degrade under root heuristic miscalibration, how robust is the method to noisy or biased heuristic estimates? A stress test with deliberately perturbed heuristic signals would clarify whether the gains depend on well calibrated estimates.

4. The Sokoban results are informative, but the paper does not clearly characterize when rerooting helps and when it does not. A more concrete analysis of the state space properties associated with effective or ineffective rerooting would strengthen the discussion.

**Limitations:**

Yes

**Strengths And Weaknesses:**

**Strengths.**

Replacing explicit subgoal generation with soft rerooting weights is conceptually elegant and appears more scalable than approaches that require reconstruction of explicit future states.

The theoretical and empirical components are well aligned. The additive rerooter result is not merely a formal addition, but directly motivates the hybrid rerooter evaluated in the experiments. This connection gives the paper a coherent structure and strengthens the overall argument.

The experimental comparisons are well chosen. In particular, the inclusion of subgoal based baselines helps isolate the practical benefit of avoiding explicit subgoal reconstruction and separate subgoal generation networks, rather than only showing gains over generic search or reinforcement learning baselines.

The BoulderDash complexity scaling experiment is a notable strength. It supports the claim that rerooting avoids part of the brittleness that arises when increasingly cluttered future states must be reconstructed. This is among the most convincing pieces of evidence for the proposed approach in more complex settings.

The discussion of Sokoban is useful and improves credibility, since it shows that the clustering signal is not uniformly beneficial across domains and that the method’s advantages are not overstated.

**Weaknesses.**

The empirical story places more emphasis on online training efficiency than on final policy quality. The held out test section is useful, but the practical claim would be stronger with a more systematic comparison under matched test time budgets. That would make it easier to assess whether the gains persist in the setting most relevant for downstream use.

The clustering based rerooter also introduces several practical design choices, including graph construction, clustering schedule, and rerooting update frequency, yet robustness to these choices is not explored in sufficient detail. As a result, stability outside the reported configurations remains somewhat unclear.

The heuristic rerooter appears somewhat fragile, a point the paper partly acknowledges. Normalization by the root heuristic is intuitive, but it may become brittle when the root estimate is poor. This issue merits a clearer stress test, since the current discussion identifies the concern without fully characterizing its practical importance.

The practical overhead of online clustering is discussed, but the cost is not broken down in enough detail across domains. Since scalability is part of the method’s value proposition, this accounting matters. It would be useful to clarify when the additional online computation remains minor and when it begins to offset the practical benefit of rerooting.

The baseline set is solid, but the case would be stronger with broader comparisons to other hierarchical or structure exploiting search baselines where feasible. The current evaluation is meaningful, but a wider set of comparisons would help position the contribution more clearly within the broader search literature.

---

> ### Author Rebuttal · Authors · 2026-03-29
>
> Thank you for your helpful comments and feedback.
>
> **Question 1**:
> We agree that a clearer sensitivity discussion is useful.
> For the clustering-based rerooter, the main clustering-specific hyperparameters are the update factor $\gamma$, which determines how often clustering is recomputed, and the hierarchy level $k$, which determines the cluster graph used for coloring.
> Appendices E and F were intended to study exactly these choices.
> Appendix F shows that varying $\gamma$ yields a smooth tradeoff rather than brittle tuning sensitivity, where a smaller $\gamma$ recomputes clusters more often and gives slightly stronger guidance at lower throughput, while a larger $\gamma$ improves throughput but uses stale cluster assignments more often.
> Performance is relatively stable across the tested range, except at the most extreme setting, which suggests the method is not highly sensitive around our chosen $\gamma=1.2$.
> Appendix E shows a similar pattern for $k$, where the best performance comes from an intermediate level rather than either extreme.
>
> **Question 2**:
> We agree that the test-time comparison is incomplete and will add average wall-clock time to the held-out test table.
> The result is nuanced: in some domains (e.g., TSP), rerooting overhead is negligible and the lower search effort is enough to match or improve wall-clock time, while in Sokoban the added bookkeeping is visible and rerooting is slower than LTS at test time in absolute seconds.
> In other domains not examined in this paper (e.g., theorem proving, protein folding, LLM thinking modes), the cost of bookkeeping of the search is negligible compared to the cost of simulating the environment or calling a large neural network. In such cases, the clustering algorithm is unlikely to be the bottleneck.
> We therefore do not claim universal test-time speedups.
>
> That said, our main contribution is about training efficiency and scalability, which is also the practically expensive stage.
> The time-based training curves already show that the rerooting methods make faster progress in wall-clock time, not only in expansions.
> By contrast, once training is complete, all methods act very quickly at test time, so small differences of a second or less are typically much less important to the end user than the overall cost of training.
> We will revise the paper to make this distinction explicit and to report test-time wall-clock numbers alongside the current results.
>
> **BoulderDash**
> |Algorithm | Expansions | Sol Cost | Time (s) |
> |---|---|---|---|
> |WA*|428.59|67.94|1.07|
> |LTS|171,909.74|71.62|210.68|
> |LTS$(\pi^{SG})$|209.57|83.16|1.59|
> |PHS$(\pi^{SG})$|359.86|86.61|2.70|
> $\sqrt{LTS}\text{-L}$|130.99|72.33|0.64|
> $\sqrt{LTS}\text{-H}$|79.65|69.80|0.49|
> $\sqrt{LTS}\text{-LH}$|96.33|70.02|0.48|
>
> **CraftWorld**
> |Algorithm | Expansions | Sol Cost | Time (s) |
> |---|---|---|---|
> |WA*| 1,888.35|122.16|3.62|
> |LTS|383,508.02|116.99|698.34|
> |LTS$(\pi^{SG})$|333,022.49|186.58|861.36|
> |PHS$(\pi^{SG})$|1,413.00|126.01|8.67|
> $\sqrt{LTS}\text{-L}$|5,601.52|174.22|26.27|
> $\sqrt{LTS}\text{-H}$|1,525.99|135.32|3.94|
> $\sqrt{LTS}\text{-LH}$|1,091.95|131.20|2.82|
>
> **Sokoban**
> |Algorithm | Expansions | Sol Cost | Time (s) |
> |---|---|---|---|
> |WA*|1,994.85|32.26|0.79|
> |LTS|2,076.52|36.46|0.89|
> |LTS$(\pi^{SG})$|2,010.88|35.82|1.98|
> |PHS$(\pi^{SG})$|1,630.60|34.66|1.56|
> $\sqrt{LTS}\text{-L}$|2,646.97|40.13|1.58|
> $\sqrt{LTS}\text{-H}$|4,044.40|36.84|1.95|
> $\sqrt{LTS}\text{-LH}$|3,635.07|36.77|2.25|
>
> **TSP**
> |Algorithm | Expansions | Sol Cost | Time (s) |
> |---|---|---|---|
> |WA*|197.28|36.04|0.51|
> |LTS|53.40|36.38|0.34|
> |LTS$(\pi^{SG})$|82.55|36.49|0.71|
> |PHS$(\pi^{SG})$|46.31|36.39|0.49|
> $\sqrt{LTS}\text{-L}$|90.64|36.39|0.44|
> $\sqrt{LTS}\text{-H}$|50.33|36.60|0.30|
> $\sqrt{LTS}\text{-LH}$|52.53|36.58|0.30|
>
> **Question 3**:
> We agree that robustness to heuristic miscalibration is important.
> To test this directly, we reran TSP (the only environment where we observed a noticeable training slowdown plausibly linked to heuristic miscalibration) using the heuristic-only rerooter with deliberately perturbing the rerooter signal by adding Gaussian noise $\epsilon \sim \mathcal{N}(0, 0.1)$ to the rerooting term before clipping.
> Under this perturbation, the slowdown disappeared.
> At test time, with the same noise still applied, the average number of expansions and solution length were 52.83 and 36.58, respectively, indicating that performance remains stable under moderate noise.
> This suggests that the method is not overly dependent on well-calibrated heuristic estimates, at least in TSP.
> We believe this is partly because the overall procedure is not purely greedy: batched inference allows multiple open-list nodes to be evaluated together, and batched bootstrap updates reduce sensitivity to any single misranked trajectory.
> We will clarify this point in the revision.
>
> **Question 4**: See the answer to y2y6's Question 1

---

> > ### Author Rebuttal · Reviewer_wpxn · 2026-04-04
> >
> > The rebuttal is appreciated. After reading the response and considering the other reviews, the questions raised in this review appear resolved. The overall assessment remains unchanged.

---

### Decision · Program_Chairs · 2026-04-30

**Decision:**

Accept (regular)

**Comment:**

The reviewers found this paper technically solid and appreciated its clear contribution: replacing explicit subgoal generation with learned rerooters based on structural signals for scalable policy-guided tree search. The combination of theory and experiments was seen as coherent, and the BoulderDash scaling results were particularly convincing.

The main concerns were about the completeness of the empirical characterization and the wording of some claims. The rebuttal addressed these points well by clarifying the baselines, refining the claims, and adding useful quantitative analysis and limitations. Overall, the remaining issues were not viewed as fundamental.

This paper fills an important gap in the rerooting literature and offers a promising, well-supported direction for scalable search. I therefore recommend acceptance.